# *Burkholderia pseudomallei* BipD modulates host mitophagy to evade killing

Dongqi Nan [1,6], Chenglong Rao [1,6], Zhiheng Tang [2,6], Wenbo Yang [1,6], Pan Wu[1], Jiangao Chen[1], Yupei Xia[1], Jingmin Yan[1], Wenzheng Liu[1], Ziyuan Zhang[1], Zhiqiang Hu[1], Hai Chen[3], Yaling Liao[1], Xuhu Mao [1,4] ✉, Xiaoyun Liu [2] ✉, Quanming Zou[5] ✉ & Qian Li [1,4] ✉

Mitophagy is critical for mitochondrial quality control and function to clear damaged mitochondria. Here, we found that *Burkholderia pseudomallei* maneuvered host mitophagy for its intracellular survival through the type III secretion system needle tip protein BipD. We identified BipD, interacting with BTB-containing proteins KLHL9 and KLHL13 by binding to the Back and Kelch domains, recruited NEDD8 family RING E3 ligase CUL3 in response to *B. pseudomallei* infection. Although evidently not involved in regulation of infectious diseases, KLHL9/KLHL13/CUL3 E3 ligase complex was essential for BipD-dependent ubiquitination of mitochondria in mouse macrophages. Mechanistically, we discovered the inner mitochondrial membrane IMMT via host ubiquitome profiling as a substrate of KLHL9/KLHL13/CUL3 complex. Notably, K63-linked ubiquitination of IMMT K211 was required for initiating host mitophagy, thereby reducing mitochondrial ROS production. Here, we show a unique mechanism used by bacterial pathogens that hijacks host mitophagy for their survival.

Mitochondria, complex and dynamic double-membraned structures, have a crucial role in controlling a wide range of functions across various biological processes and the immune response of the host to bacterial infections[1]. Mitophagy, vital for maintaining mitochondrial quality and functionality, is activated due to excess damage to mitochondria or oxidative stress, subsequently recognized by the autophagosome via engagement with cytoplasmic microtubule-related protein 1 light chain 3 (LC3), followed by autophagic degradation. This process constrains the buildup of damaged mitochondrial DNA (mtDNA) and reactive oxygen species (mtROS) in host cells by removing the damaged mitochondria. Plenty of studies have proved that mitophagy is manipulated by a variety of pathogens and their

components to enhance their intracellular survival or escape the host's immunity[2,3], including pathogens human parainfluenza virus type 3, hepatitis C virus, human herpesvirus 8, and *Listeria monocytogenes*[4–8]. Typically, molecular mechanisms of mitochondrial turnover by mitophagy have been classified into three main pathways, the PTEN-induced kinase 1 (PINK1)/Parkin pathway, and the adapter proteins or receptors-mediated pathways in either ubiquitin (Ub)-dependent or independent[2,9]. While Parkin is a significant regulator of mitophagy, current research shows that mitophagy can still be triggered in the absence of Parkin. Recent findings indicate that mitochondria-localized viral interferon regulatory factor 1 facilitates mitochondrial clearance and viral replication during HHV-8 infection by activating the

[1]Department of Clinical Microbiology and Immunology, College of Pharmacy and Medical Laboratory, Army Medical University (Third Military Medical University), Chongqing, China. [2]Department of Microbiology and Infectious Disease Center, NHC Key Laboratory of Medical Immunology, School of Basic Medical Sciences, Peking University Health Science Center, Beijing, China. [3]Sanya People's Hospital, Sanya, China. [4]State Key Laboratory of Trauma and Chemical Poisoning, Army Medical University (Third Military Medical University), Chongqing, China. [5]Department of Microbiology and Biochemical Pharmacy, College of Pharmacy and Laboratory Medicine, Army Medical University (Third Military Medical University), Chongqing, China. [6]These authors contributed equally: Dongqi Nan, Chenglong Rao, Zhiheng Tang, Wenbo Yang. ✉e-mail: maoxh2012@hotmail.com; xiaoyun.liu@bjmu.edu.cn; qmzou@tmmu.edu.cn; liqianjane@163.com

receptor NIX-mediated mitophagy[10]. Moreover, the intracellular pathogen *L. monocytogenes* induces mitophagy through the LC3 receptor NOD-like receptor X1 (NLRX1) mediated by its virulence factor listeriolysin O[8]. Nevertheless, whether other bacterial pathogens regulate mitochondrial homeostasis remains largely elusive.

Profound comprehension of the molecular mechanisms underpinning innate immune reactions to bacterial pathogens is crucial for developing effective treatment strategies against infections. *Burkholderia pseudomallei*, the etiological agent of melioidosis, was initially identified in Rangoon by Whitmore and Krishnaswami in 1911[11]. Infection with *B. pseudomallei* is usually caused by physical injury, consumption, or inhalation of aerosolized bacteria, with higher incidences during the wet season[12]. The global underdiagnosis of melioidosis is largely attributed to the nonspecific nature of its clinical symptoms and the lack of specialized microbiological laboratories, leading to mortality rates as high as 40%[13–15]. Moreover, emerging data suggests that the severity and prognosis of melioidosis largely depend on the specific risk factors, bacterial virulence factors, as well as the bacterial load and strain[16,17]. Even though the strategies of *B. pseudomallei* for intracellular survival are being gradually uncovered, there is currently no available vaccine for melioidosis[18–20], making continued research into the pathogenesis of utmost importance.

Herein, a model system using *B. pseudomallei* was utilized to demonstrate that bacterial pathogens inside the cell instigate the mitophagy of immune cells of the host to facilitate the survival. In our efforts to search for potential mitophagy inducer from *B. pseudomallei*, we found BipD, previously recognized as the type three secretion system (T3SS) needle tip protein, could induce mitophagy after *B. pseudomallei* infection. Rather than depending on PINK1/Parkin axis or mitophagy receptors, BipD-mediated mitophagy induction in host cells by hijacking KLHL9/KLHL13/CUL3 E3 ligase complex in a Ub-dependent manner. Furthermore, we identified IMMT, as a substrate of KLHL9/KLHL13/CUL3 complex, was undergoing K63-linked ubiquitination and autophagic degradation, which led to the clearance of mtROS and facilitated bacterial survival. In this work, the finding illuminates a previously unexplored aspect of the complicated interaction between pathogen and host in innate immunity during infections.

## Results

### *B. pseudomallei* induces mitophagy in macrophages

To address mechanisms of mitochondrial quality control upon *B. pseudomallei* infection, we measured the mitochondrial membrane potential (MMP, Δψm) and found *B. pseudomallei* infection resulted in a significant loss of Δψm as determined by using flow cytometric (Supplementary Fig. 1a, b) and microscopic analyses (Supplementary Fig. 1c). Next, we demonstrated that mitophagy, but not mitocytosis[21] occurred in response to the mitochondrial damage induced by *B. pseudomallei* (Supplementary Fig. 1c–e). To further confirm these observations, we analyzed the levels of mitochondrial DNA (mtDNA) and protein (HSP60 and TIM23) in RAW264.7 cells. The experiments revealed a significant reduction in mtDNA levels following infection (Fig. 1a, b) and the amount of HSP60 and TIM23 (Fig. 1c, d), in a manner dependent on both the time and multiplicity of infection (MOI). Additionally, an increase in the colocalization of mitochondria with LC3 was observed (Fig. 1e, f). Further investigation through transmission electron microscopy (TEM) evidenced that mitochondria were enclosed within double membrane compartments (autophagosomes or autolysosomes) in infected cells, reminiscent of CCCP treatment (Fig. 1g, h). We further stained the lysosomes with Lysotracker and found significantly increased colocalization of mitochondria and lysosomes as well (Fig. 1i, j), indicating potential degradation of mitochondria eventually by lysosomes. Subsequently, it was ascertained that *B. pseudomallei* infection could trigger a similar response in mouse peritoneal macrophages (PMs), which caused a decrease of

mtDNA levels and an obvious degradation of HSP60 and TIM23 (Fig. 1k, l). Additionally, more mitochondria enclosed by autophagosomes/autolysosomes were visible through TEM after *B. pseudomallei* infection (Fig. 1m). Taken together, these data indicate that *B. pseudomallei* indefection induces mitophagy in host cells.

Next, we sought to determine whether conventional autophagy is involved in *B. pseudomallei*-induced mitophagy. Both *B. pseudomallei*- and CCCP-induced decreases in the levels of mtDNA and mitochondrial proteins were blocked in *Atg5* knockdown macrophages (Supplementary Fig. 2a–d). Furthermore, the number of red mt-Keima puncta was significantly reduced when *Atg5* was knocked down, indicating a declined level of mitolysosomes (Supplementary Fig. 2e). Similarly, as shown in Supplementary Fig. 2f–h, the decrease in mtDNA induced by *B. pseudomallei* infection was also suppressed after siRNA-mediated knockdown of other autophagy associated genes (*Atg7* and *FIP200*). Taken together, these results confirm that *B. pseudomallei* infection has the capability to induce mitophagy in mouse macrophages, which is dependent on the conventional autophagy.

### *B. pseudomallei* hijacks host mitophagy for its survival by decreasing mtROS

In the subsequent phase of the research, the focus was shifted to discern the influence of mitophagy on intracellular survival of *B. pseudomallei*. We found that the intracellular titer of *B. pseudomallei* significantly increased under CCCP treatment in RAW264.7 cells and mouse PMs, respectively (Fig. 2a, b). Further examination revealed that under CCCP treatment, the loads of *B. pseudomallei* were elevated in both liver and spleen of infected mice (Fig. 2c). Consequently, these findings strongly suggest that the stimulation of mitophagy may enhance intracellular survival of *B. pseudomallei* in vivo and in vitro.

Subsequently, we focused on understanding mechanisms by which mitophagy supports *B. pseudomallei* survival. Mitochondrial reactive oxygen species (mtROS) has been previously recognized as assuming a vital part in the immunoreaction to intracellular bacterial infections, which acts as a proinflammatory signal or produces hydrogen peroxide ($H_2O_2$) to promote the killing of intracellular bacteria[1,22]. Therefore, we examined whether mtROS production in macrophages is essential for the efficient clearance of *B. pseudomallei*. As shown in Fig. 2d, we observed that *B. pseudomallei* infection induced a significant increase of mtROS, but CCCP treatment attenuated the increase in macrophages. Similar results were obtained in PMs (Fig. 2i). Additionally, through the bacterial quantification using immunofluorescence staining, it was inferred that *B. pseudomallei* achieved a higher intracellular titer when coinciding with lower mtROS levels in RAW264.7 cells (Fig. 2e–g). In order to further validate the influence of mtROS on intracellular *B. pseudomallei* eradication, the mtROS scavenger MitoTEMPO was utilized. As illustrated in Fig. 2h, j, MitoTEMPO treatment markedly facilitated the survival of *B. pseudomallei* within macrophages. Consistently, MitoTEMPO treatment could also increase the intracellular titer of *B. pseudomallei* in liver and spleen of mice (Fig. 2k, l). Collectively, these results suggest that *B. pseudomallei* infection induces mitophagy to decrease the level of mtROS in host cells in order to enhance its intracellular survival.

### BipD induced host mitophagy post *B. pseudomallei* infection

Proceeding with the investigation to delineate the molecular pathways involved in *B. pseudomallei*-induced mitophagy, an initial examination revealed that *B. pseudomallei* infection led to the PINK1 accumulation, without corresponding involvement of Parkin in mitochondria (Supplementary Fig. 3a–c). Furthermore, *B. pseudomallei*-induced mitophagy was not affected when *Pink1* or *Parkin* was knockdown (Supplementary Fig. 3d–f and 3i), underlying that PINK1/Parkin pathway may not be required for *B. pseudomallei* infection. Meanwhile, we tested the role of currently reported mitophagy receptors and found that knockdown of these receptors had no effect on *B. pseudomallei*-

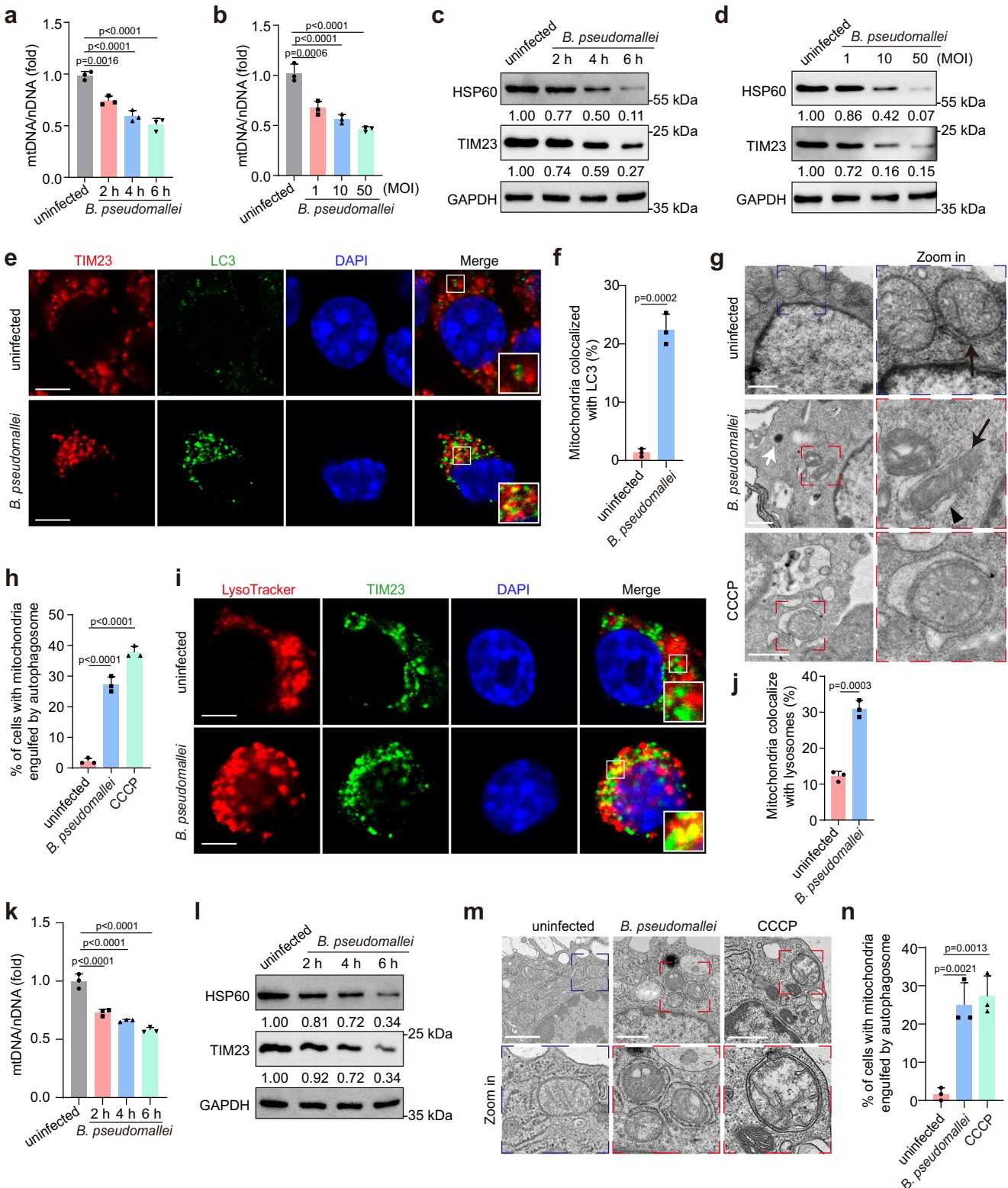

induced mitophagy, respectively (Supplementary Fig. 3j–l). Further-more, *B. pseudomallei*-induced mitochondrial ubiquitination even when Parkin was knockdown (Supplementary Fig. 3g, h). Thus, it is likely that *B. pseudomallei*-induced mitophagy is in a ubiquitin-dependent manner.

Recent evidence has revealed that pathogens maneuver the host immune response to facilitate their own survival by bacterial effectors[23,24]. Nucleic acid or heat-killed *B. pseudomallei* was used to treat RAW264.7 cells. It was found that neither of them could induce mitophagy in host cells (Supplementary Fig. 4a–c), implying that induction of mitophagy was actively modulated by *B. pseudomallei*. Thus, we hypothesized that there might be a potential effector to mediate *B. pseudomallei*-induced mitophagy. Previous studies have shown that pathogen proteins containing LC3-interacting region (LIR) motif can act as the adapter protein of selective autophagy, mediating the autophagosome to wrap the damaged substrate[25–27]. Given that

**Fig. 1 | *B. pseudomallei* infection induces mitophagy in mouse macrophages.**
**a, b** Quantification of mtDNA/nDNA levels in RAW264.7 cells upon infection of *B. pseudomallei* by qPCR analysis. Cells were *B. pseudomallei*-infected for different time points (at an MOI of 10) or at the indicated MOI for 4 h. MOI, multiplicity of infection. **c, d** Detection of the mitochondrial proteins HSP60 and TIM23 in *B. pseudomallei*-infected RAW264.7 cells. Cells were under treatment in (**a**) and (**b**), respectively. **e** Typical immunofluorescence photos of colocalizing TIM23 and LC3 in infected RAW264.7 cells using confocal microscopy. Scale bar, 5 µm. **f** Quantification of LC3-colocalized TIM23 represented in (**e**). 50 cells in 10 fields were calculated in all groups from three distinct trials. **g** Representative electron micrographs of mitochondria engulfed by autophagosome in infected or CCCP (10 µM) treated RAW264.7 cells. White arrow indicates bacteria, black arrow indicates mitochondria, and black triangle indicates autophagosomes. Scare bar, 1 µm. (**h**) Quantification of mitochondria engulfed by autophagosome represented in (**g**).

20 cells were calculated in each group from three distinct trials. **i** Typical images show TIM23 colocalized with the lysosome marker Lysotracker in infected RAW264.7 cells. Scale bar, 5 µm. **j** Quantifying TIM23 colocalized with lysosomes represented in (**i**). 50 cells in 10 fields were calculated in all groups from three distinct trials. **k, l** Measurement of the mtDNA/nDNA and HSP60 and TIM23 levels in *B. pseudomallei*-infected mouse peritoneal macrophages (PMs) for preset time points. **m** Representative TEM results of mitochondria engulfed by autophagosome in mouse PMs with *B. pseudomallei* infection or CCCP treatment (10 µM). Scare bar, 1 µm. **n** Quantification of mitochondria engulfed by autophagosome represented in (**m**). 20 cells were calculated in all groups from three distinct trials. The above data indicate means ± SD for three independent trials. One-way ANOVA followed by Tukey post hoc test (**a, b, h, k, n**) and two-tailed Student's *t*-test (**f, j**) were used for data analysis. *n* = 3 in each group (**a, b, k**). Source data were provided as a Source Data file.

T3SS is a critical virulence factor in *B. pseudomallei*, we analyzed all the reported proteins of T3SS and found eight proteins were predicted to contain LIR motif by iLIR and hAIM with a PSSM score of 18 or more (Supplementary Data 1). Then, we found no significant changes of Δψm were observed under overexpression of these 8 proteins, respectively (Supplementary Fig. 5a). Nonetheless, with the CCCP pretreatment, we found that BPSS1401, BPSS1537 and BipD appeared to play a role in mitophagy induction, and BipD had the most significant effect on regulating the response (Supplementary Fig. 5b–e).

To corroborate these findings, an analysis was conducted to determine the secretion of BipD into host cells through the T3SS. Thus, β-lactamase reporter system assay demonstrated that BopA[28], as a T3SS effector protein, or BipD-TEM fusion protein was efficiently secreted into RAW264.7 cells from wild-type (WT) *B. pseudomallei*, but not its T3SS-deficient Δ*bsaZ* mutant (Supplementary Fig. 6a–c). Next, we found that BipD overexpression induced a marked reduction in both mtDNA and mitochondrial protein levels under the CCCP treatment (Fig. 3a, b). Congruent results were also discerned in the increased colocalization of mitochondria with LC3 (Fig. 3c, d). Then, we considered that BipD-mediated mitophagy occurrence might delete the CCCP-induced damaged mitochondria, and the Δψm could be recovered over time. In order to experimentally verify this hypothesis, control and BipD-overexpressing cells were initially exposed to CCCP for a duration of 1 h, followed by a subsequent 4-h culture in a standard medium. The recovery of Δψm in BipD-overexpressing cells was much stronger than that in control ones, with initially similar Δψm after 1 h CCCP treatment (Fig. 3e, f). Thus, these results implied that mediation of mitophagy by BipD is necessary but not sufficient for mitophagy induction.

Furthermore, TEM images showed that mitochondria existed structural changes such as vacuoles, swelling and severed mitochondrial membranes in either *B. pseudomallei* WT or Δ*bipD* infected PMs (Fig. 3g). Especially, infection with Δ*bipD* results in a more distinctly structural damage of mitochondria without double-/multi-membrane compartments (autophagosomes/autolysosomes) encompassed compared with the WT group. Consistently, the colocalization of mitochondria with GPF-LC3 and recovery of Δψm were much lower (Fig. 3h, i), and less reduction in the mtDNA and protein levels of HSP60 and TIM23 under Δ*bipD* infection (Fig. 3j, k). Furthermore, measuring bacterial replication revealed that *B. pseudomallei* Δ*bipD* grew less efficiently in PMs than WT strain, and this phenotype could be restored by Δ*bipD*:*bipD* (Fig. 3l). Of note, BopA has been proved to facilitates bacterial evasion of autophagy by interfering with LC3-associated phagocytosis (LAP) in *B. pseudomallei*-infected RAW264.7 cells[28,29]. To further confirm whether BopA is involved in the BipD-mediated mitophagy initiation, we found that infection with *B. pseudomallei* WT and Δ*bopA* resulted in similar decreases in the levels of mtDNA, mitochondria proteins, and Δψm changes, respectively (Supplementary Fig. 7a–c). Moreover, ectopic expression of BipD in cells infected with Δ*bipD* strain could rescue the decrease in the mtDNA,

and promote the degradation of accumulated mitochondrial proteins, while expression of BopA showed no differences under Δ*bopA* infection (Supplementary Fig. 7d, e). Similar effect was observed in the colocalization of mitochondria marker HSP60 and LC3 (Supplementary Fig. 7f, g). Taken together, these results suggested that BipD is essential to mediate the induction of host mitophagy after *B. pseudomallei* infection, while BopA may not be involved in this process.

## BipD interacts with the BTB-containing adapters KLHL9 and KLHL13

Considering its predicted LIR motif, we firstly examined whether BipD interacts with LC3 by coimmunoprecipitation (co-IP) analysis. We found that BipD could not interact with LC3 compared to the negative control of *Salmonella* Type III effector SopD2[30] (Supplementary Fig. 5f). Further exploration to pinpoint the underlying substrate(s) of BipD was conducted through the utilization of mass spectrometry (MS) to detect proteins capable of binding to BipD. MS analysis identified a total of 63 upregulated proteins (upregulated over 2-fold change, and P value <0.05), and the top 3 targets interacting with BipD included the KIF11, KLHL9, and KLHL13 (Fig. 4a and Supplementary Data 2). Accumulating evidence has indicated that several Kelch-like (KLHL) proteins are required for the recognition between the E3 ligase CUL3 and its substrate, and proved involved in ubiquitination[31]. Therefore, we speculated that BipD might interact with KLHL9 and KLHL13. Then, we constructed the eukaryotic expression vectors of His-tagged KLHL9 and KLHL13, and co-transfected them in HEK293T cells with Flag-tagged BipD, respectively. As shown in Fig. 4b, c, co-IP assays validated the interaction of Flag-tagged BipD with His-tagged KLHL9 or KLHL13, respectively. Consistently, GST pulldown assays further confirmed the direct association between purified GST-BipD and His-tagged KLHL9 and KLHL13, respectively (Fig. 4d, e). Then, in order to map the BipD-binding site on KLHL9 or KLHL13, we constructed different deletion mutants of KLHL9 and KLHL13 with Xpress tag according to the reported domains, respectively (Fig. 4f). As depicted in Fig. 4g, co-IP analysis demonstrated that only the M5 and M10 fragment of KLHL9 and KLHL13 (containing the Back and six Kelch repeat domains) interacted with BipD, respectively, which is similar to the full-length KLHL9 or KLHL13, suggesting that the structure including both Back and six Kelch repeat domains of KLHL9 and KLHL13 was essential for BipD binding. Collectively, these above results indicate that BipD interacts with the BTB-containing proteins KLHL9 and KLHL13 through the Back and Kelch domains.

## BipD and KLHL9/KLHL13/CUL3 complex act synergistically to ubiquitinate mitochondria by targeting IMMT

Recent studies have underscored KLHL9 and KLHL13 involvement in ubiquitination via CUL3. To confirm the binding of CUL3 with KLHL9 and KLHL13 after *B. pseudomallei* infection, we found that BipD could interact with KLHL9/KLHL13/CUL3 complex by co-IP analysis in HEK293T cells (Fig. 5a). Meanwhile, observations by confocal

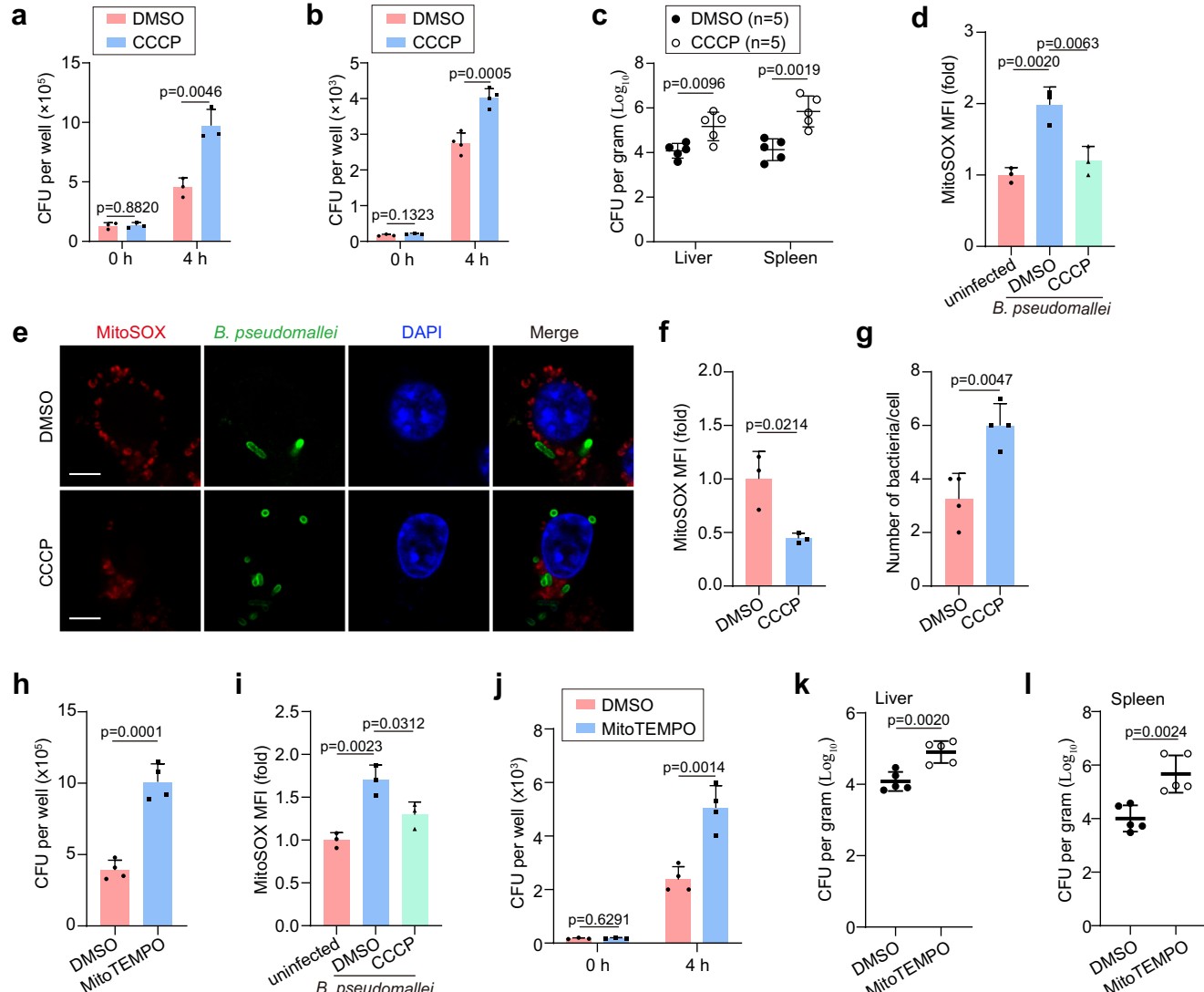

**Fig. 2 | B. pseudomallei initiates mitophagy to decrease mtROS for the intracellular survival in vivo and in vitro. a** The influence of CCCP on the load of *B. pseudomallei* inside RAW264.7 cells. CFU, colony-forming units. **b** The impact of CCCP treatment on the intracellular bacterial load in *B. pseudomallei*-infected mouse PMs pretreated by 10 μM CCCP. 0 h, $n = 3$. 4 h, $n = 4$. **c** Bacterial loads in liver and spleen of BALB/c mice after 3 day post-infection. Mouse models were intraperitoneal (i.p.) infection with $4 \times 10^5$ CFU *B. pseudomallei* at day 1, and then injected i.p. with DMSO or CCCP (5 mg/kg body mass) at day 2, respectively. **d** Measurement of the impact of CCCP treatment on levels of mtROS through flow cytometry in RAW264.7 cells. **e** Typical immunofluorescence photos of RAW264.7 with MitoSOX staining following *B. pseudomallei* infection. Scare bar, 5 μm. **f, g** Quantification of MitoSOX MFI and the bacterial count each cell in (**e**). 50 cells in 10 fields were calculated in all groups from three separate trials. **h** The effect of DMSO and MitoTEMPO (0.5 mM) on *B. pseudomallei* load inside RAW264.7 cells. **i** Measurement of the effect of DMSO and CCCP (10 μM) on the mtROS production in *B. pseudomallei*-infected mouse PMs. **j** The effect of DMSO and MitoTEMPO (0.5 mM) on intracellular *B. pseudomallei* load in PMs. DMSO, $n = 3$. MitoTEMPO, $n = 4$. **k, l** Mice models were i.p. infected with $4 \times 10^5$ CFU *B. pseudomallei* at day 1, and then injected i.p. with DMSO or MitoTEMPO (100 mg/kg body mass) at day 2. The above data indicate means ± SD for three or more independent trials. One-way ANOVA followed by Tukey post hoc test (**d, i**) and two-tailed Student's *t*-test (**a–c, f–h, j–l**) were used for data analysis. $n = 3$ in each group (**a, d, i**), $n = 4$ in each group (**g, h**), $n = 5$ in each group (**c, k, l**). Source data were provided as a Source Data file.

microscopy showed that the colocalization of mitochondria with ubiquitin were decreased when *Klhl9*, *Klhl13* or *Cul3* was knockdown, respectively (Fig. 5b, c). Consistently, we analyzed the ubiquitination levels of cell lysate and mitochondria, and further verified that mitochondrial ubiquitination was declined by *Cul3*, *Klhl9* or *Klhl13* knockdown, while no significant changes in cell lysates (Fig. 5d).

Then, given that KLHL9 and KLHL13 function as substrate-specific adapter by CUL3-based ubiquitination modification, we aimed to explore the detailed mitochondrial substrate(s) mediated by BipD. As shown in Fig. 5e and Supplementary Data 3, we carried out ubiquitin proteomics analyses and 23 mitochondrial related proteins were measured. Particularly, TIMM50 and IMMT (Mitofilin), as the mitochondrial membrane proteins, were further verified by co-IP analysis,

respectively, and only ubiquitination of IMMT was proved to be increased after *B. pseudomallei* infection (Supplementary Fig. 8a, b). Next, we further confirmed that CUL3 were colocalized with mitochondrial HSP60 under ectopic expression of BipD by fluorescence microscopy (Supplementary Fig. 8c). Of note, we found a discontinuous fluorescence signal of TOMM20 and the colocalization of Ub and IMMT in response to *B. pseudomallei* infection (Supplementary Fig. 8d, e), indicating that outer mitochondrial membrane (OMM) severing may contribute to a possibility for IMMT exposure and ubiquitination following infection.

Besides, SQSTM1, as an adapter implicated in Ub-dependent mitophagy, was also identified to show evidently increased ubiquitination (Fig. 5e). Thus, we reasoned that SQSTM1 may be involved in the

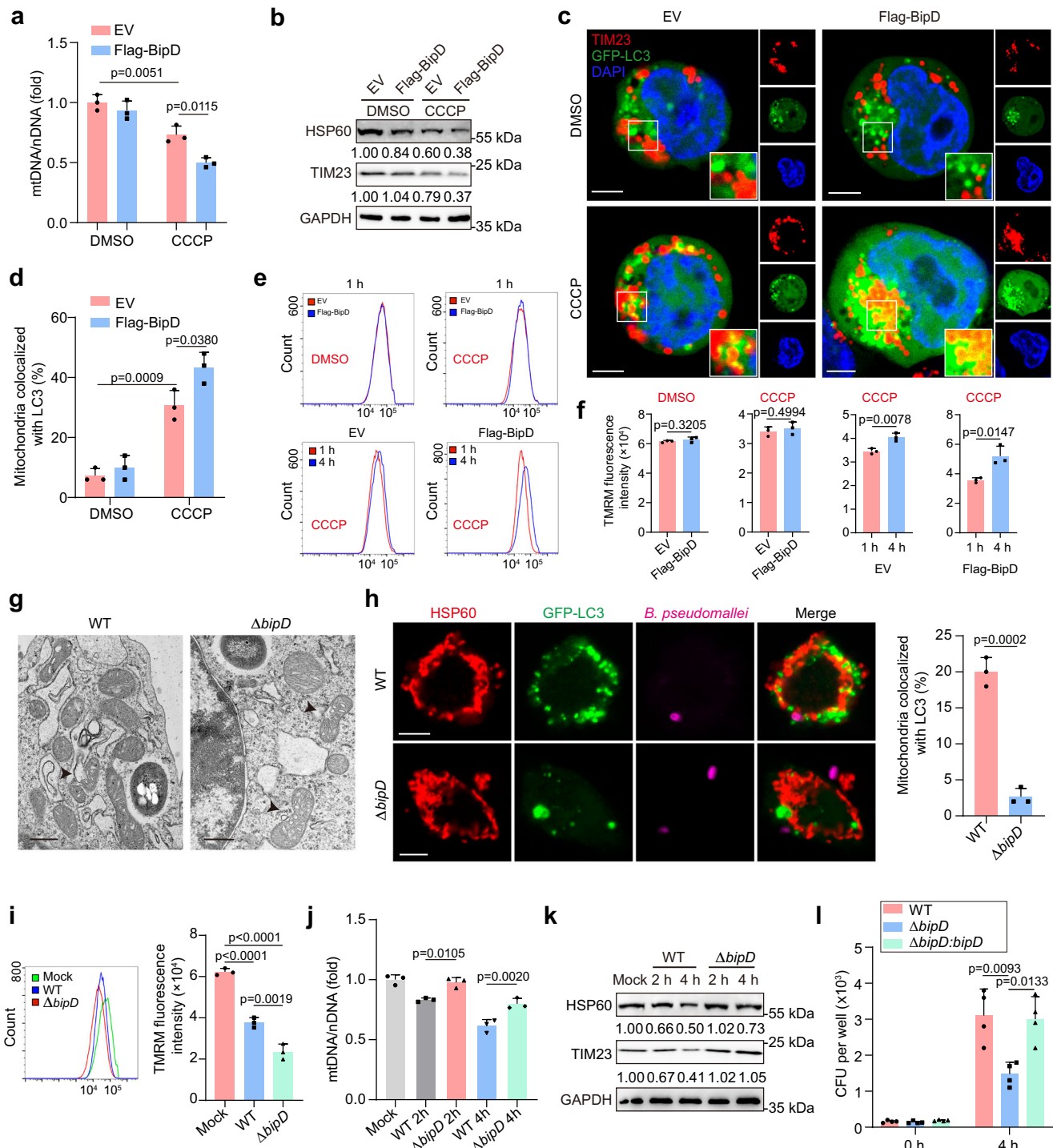

**Fig. 3 | *B. pseudomallei* infection induced host mitophagy depending on BipD.**
**a** Quantification of mtDNA/nDNA levels in HEK293T cells. Cells were transfected with empty vector (EV) or Flag-BipD plasmid and then treated with DMSO or CCCP. **b** Western blot analyzed protein levels of HSP60 and TIM23 in HEK293T cells under treatment in (**a**). **c** Colocalization of TIM23 and GFP-LC3 was visualized in HEK293T cells using confocal microscopy. Cells were transfected with GFP-LC3 plasmid and subsequent treatment in (**a**). Scare bar, 5 μm. **d** Quantifying TIM23 colocalized with GFP-LC3 represented in (**c**). Data were from 3 independent experiments with 50 cells in each group. **e** Measurement of mitochondrial membrane potential (MMP, Δψm) in HEK293T cells determined by staining of TMRM. Cells were transfected with EV or Flag-BipD plasmids, treated with DMSO or CCCP for 1 h, and subsequently maintained in a standard medium for an another 3 h. **f** Determined the fluorescence intensity of TMRM. Trials were separately repeated

thrice. **g** Typical TEM photos of mitochondrial morphology in mouse PMs infected with *B. pseudomallei* WT or Δ*bipD*. Black arrows indicate severed mitochondrial membranes. Scale bar, 0.5 μm. **h** Observation and quantification of colocalizing HSP60 and GFP-LC3 in *B. pseudomallei* WT or Δ*bipD* infected mouse PMs. Scale bar, 5 μm. Data were from 3 independent experiments with 50 cells in each group. **i** Δψm was detected by FACS analysis in PMs. Cells were treated as in (**g**). **j, k** Detection of levels of the mtDNA/nDNA and the HSP60 and TIM23 proteins in *B. pseudomallei* WT or Δ*bipD* infected PMs. **l** Intracellular *B. pseudomallei* loads in PMs infected with *B. pseudomallei* WT, Δ*bipD* or Δ*bipD:bipD*. The data indicate means ± SD for three or more independent trials. Two-way ANOVA followed by Tukey post hoc test (**a, d**), two-tailed Student's *t*-test (**f, h**) and One-way ANOVA followed by Tukey post hoc test (**i, j, l**) was used for data analysis. *n* = 3 per group (**a, d, f, h–j**), *n* = 4 per group (**l**). Source data were provided as a Source Data file.

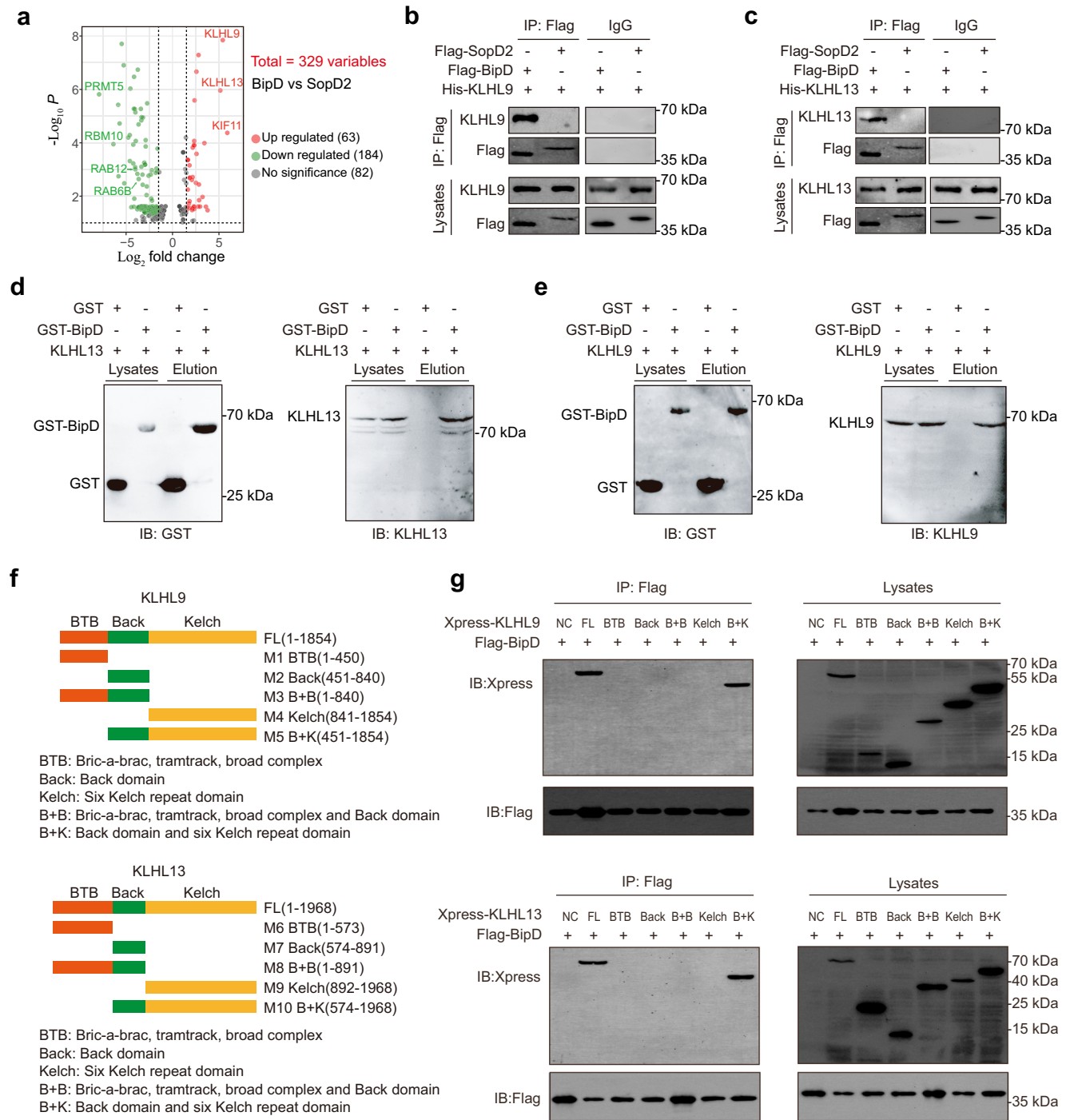

**Fig. 4 | BipD interacts with KLHL9 and KLHL13. a** Volcano plots representing genes interacted with BipD in HEK293T cells. Red and blue dots present BipD or SopD2 interacting proteins, respectively (fold change >2). Two-tailed Student's *t*-test were used for data analysis. (*n* = 4 in each group). **b, c** The interaction of BipD with KLHL9 or KLHL13 was detected by Co-IP analysis in HEK293T cells. Cells were subjected to transfection with Flag-SopD2, Flag-BipD, His-KLHL9 or His-KLHL13, respectively. **d, e** Pulldown analysis of recombinant GST-BipD with KLHL9 or KLHL13 in HEK293T cell lysates. **f** The illustration of KLHL9 and KLHL13 deletion mutants. **g** The binding site of BipD and KLHL9 or KLHL13 is required both the Back and six Kelch repeat domains, respectively. HEK293T cells were subjected to transfection with preset deletion mutant plasmids, and detected by Co-IP of anti-Flag affinity gels and then assayed with anti-Xpress or anti-Flag by Western blot. Results characteristic of three or more separate trials. Source data were provided as a Source Data file.

ubiquitinated IMMT triggered mitophagy, and found that SQSTM1 and the autophagosome marker LC3 colocalized with Ub in *B. pseudomallei*-infected RAW264.7 cells, respectively (Supplementary Fig. 9a–c), implying that ubiquitinated IMMT may provide the signal for recruiting SQSTM1 upon *B. pseudomallei* infection, mediating the initiation of host mitophagy. In addition, according to the current

evidence, the factors known to result in rupture of the outer membrane during inner membrane mediated mitophagy are Parkin[32], BAX/BAK[33], and dynamin-related protein 1 (DRP1)[34]. We therefore investigated the possibility whether the BAX/BAK or DRP1 was involved in *B. pseudomallei*-induced mitophagy. As shown in the Supplementary Fig. 9d–f, knockdown of *Drp1* obviously reversed the decline of mtDNA

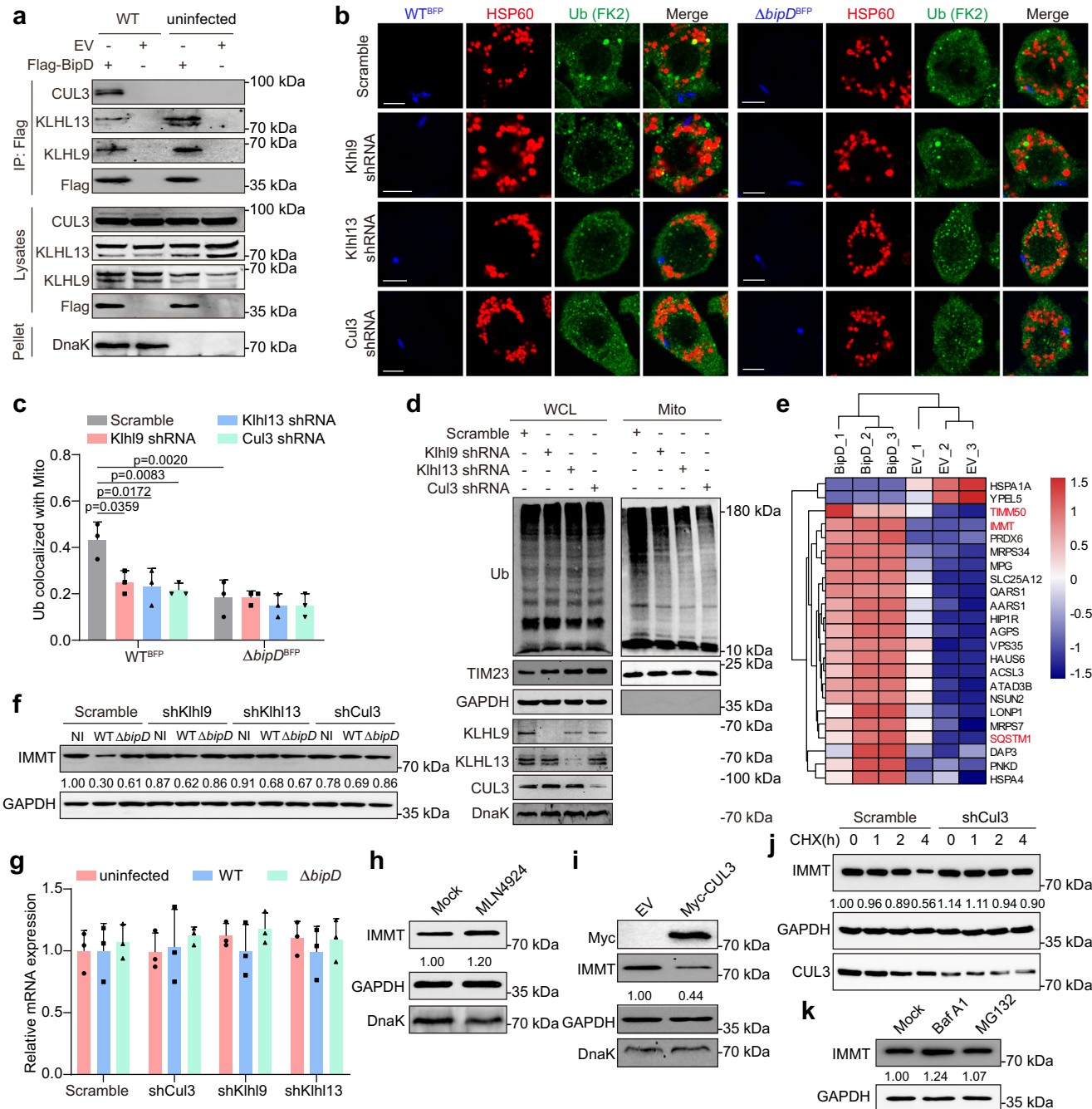

**Fig. 5 | BipD recruits KLHL9/KLHL13/CUL3 complex to ubiquitinate mitochondria by targeting IMMT. a** Detection of the interaction between BipD and KLHL9/KLHL13/CUL3 complex in HEK293T cells. Cells were transfected by Flag-BipD plasmid or EV, and uninfected or infected with *B. pseudomallei*. **b** Confocal microscopy analysis of HSP60 colocalization with ubiquitin (Ub) in *Klhl9*, *Klhl13* or *Cul3* knockdown RAW264.7 cells. Cells were infected with BFP-labeled *B. pseudomallei* WT or Δ*bipD*, and labeled for HSP60 and Ub. Scale bar, 5 μm. **c** Quantifying HSP60 colocalization with Ub in (**b**) from 3 independent experiments with 20 cells in each group. (Two-way ANOVA). **d** Measurement of ubiquitination in WCL and purified mitochondrial fractions in *B. pseudomallei*-infected *Klhl9*, *Klhl13* or *Cul3* knockdown RAW264.7 cells. Mito, mitochondria. WCL, whole cell lysates. **e** Heatmap of ubiquitin-modified substrates with a cutoff threshold of 1.5-fold difference in peptide abundance between Flag-BipD and EV transfected HEK293T cells in the presence of CCCP. **f, g** Measurement of the protein and mRNA

levels of IMMT in *Cul3*, *Klhl9* or *Klhl13* knockdown RAW264.7 cells following *B. pseudomallei* WT or Δ*bipD* infection. One-way ANOVA was used for data analysis. *n* = 3 per group. **h** MLN4924 treatment declined the degradation of IMMT in infected RAW264.7 cells. Cells were subjected to pretreatment with MLN4924 (300 nM) for 2 h, then infection for 4 h. **i** CUL3 overexpression improved the degradation of IMMT in infected RAW264.7 cells. Cells were subjected to transfection with Myc-Cul3 or EV plasmids for 24 h, and subsequent *B. pseudomallei* infection. **j** CUL3 regulates the degradation of IMMT in infected RAW264.7 cells. Cells were subjected to treatment with 20 μM cycloheximide (CHX) for the preset time points during *B. pseudomallei* infection, and then entire cell lysates were extracted and detected by Western blot. **k** RAW264.7 cells were subjected to treatment with 10 μM MG132 or 100 nM Baf A1 for 2 h, and subsequently *B. pseudomallei* infection for 4 h. The above data were from at least three separate trials and presented as means ± SD. Source data were provided as a Source Data file.

upon *B. pseudomallei* infection, while BAX and BAK were not likely required for *B. pseudomallei*-induced mitophagy. Moreover, we further observed that the colocalization of mitochondria with LC3 markedly decreased when *Drp1* was knockdown (Supplementary Fig. 9g). Similar effect was also obtained in Δψm detection (Supplementary Fig. 9h). Collectively, these results indicate that DRP1 may be crucial for *B. pseudomallei*-induced mitophagy.

To explore the roles of CUL3, KLHL9 and KLHL13 in degradation of mitochondrial substrates dependent on BipD, we found the protein level of IMMT was increased under the knockdown of *Cul3*, *Klhl9* or *Klhl13*, respectively, upon *B. pseudomallei* WT infection, whereas the effect was abrogated under Δ*bipD* infection in RAW264.7 cells (Fig. 5f). Nevertheless, the mRNA level of Immt showed no significant changes no matter when *Cul3*, *Klhl9* or *Klhl13* was knocked down after *B. pseudomallei* WT or Δ*bipD* infection (Fig. 5g). Additionally, to further confirm the effect of CUL3-based degradation after ubiquitination modification on IMMT, we examined infected RAW264.7 cells treated with MLN4294, a suppressant of neddylation, and discovered that the IMMT level was significantly elevated (Fig. 5h). Consistently, CUL3 overexpression led to a reduction in the IMMT level in infected RAW264.7 cells (Fig. 5i). Subsequent treatment of the infected cells with cycloheximide (CHX) to gauge the half-lives of IMMT unveiled that the *Cul3* knockdown extended the half-life of IMMT, thus reinforcing our previous observations (Fig. 5j). Moreover, as ubiquitinated proteins typically undergo degradation either through the ubiquitin-proteasome system or the autophagy-lysosome pathway, we sought to identify the specific degradation pathway of IMMT. By treating *B. pseudomallei*-infected RAW264.7 cells with either the proteasome inhibitor MG132 or the lysosome inhibitor bafilomycin A1 (Baf A1), we ascertained that Baf A1 could elevate the level of IMMT, whereas MG132 did not have this effect responsive to the infection of *B. pseudomallei* (Fig. 5k). Therefore, it was inferred from these results that BipD interacts with KLHL9/KLHL13/CUL3 complex to ubiquitinate mitochondrial substrate IMMT and improve its degradation via autophagy-lysosome pathway after *B. pseudomallei* infection.

## CUL3 promotes K63-linked ubiquitination of IMMT

Next, to further confirm the role of KLHL9/KLHL13/CUL3 complex in ubiquitination of IMMT, we found *CUL3* knockdown decreased the ubiquitination level of IMMT while CUL3 overexpression increased the alteration in HEK293T cells after *B. pseudomallei* infection (Fig. 6a, b). Moreover, to discern the specific ubiquitination type of IMMT triggered by CUL3, we performed UbiCRest assays using four distinct deubiquitinases (DUBs) with varying specific cleavage activation[35], including nonspecific DUBs USP21 and specific DUBs AMSH, OTUB1 and YOD1. As shown in Fig. 6c, DUBs USP21 and AMSH induced the cleavage of ubiquitin chains on IMMT, indicating that IMMT was mainly modified with K63-linked ubiquitin chains. Consistently, following co-transfection of HA-K63 only Ub mutant, Myc-CUL3 and IMMT-Flag plasmids, the level of K63-linked polyubiquitination of IMMT was remarkably increased under *B. pseudomallei* infection (Fig. 6d), suggesting that CUL3 induced the K63-linked ubiquitination and degradation via autophagy-lysosome pathway of IMMT. Furthermore, to explore the major ubiquitination sites of IMMT modified by CUL3, we used the GPS-Uber web to predict the lysine residues that might be ubiquitinated by CUL3. According to the prediction results, we constructed the Flag-tagged IMMT mutants of the top six lysine residues with high confidence and scores higher than 0.85 (Fig. 6e). Subsequently, IP assays demonstrated the ubiquitination of IMMT K211R mutant was decreased (Fig. 6f), implying that this lysine residue was crucial for the K63-linked ubiquitination of IMMT mediated by CUL3. Collectively, these insights revealed that CUL3 mediates the K63-linked ubiquitination of IMMT at the K211 lysine residue, thereby promoting degradation of IMMT via the autophagy-lysosome pathway.

## CUL3 and IMMT are required for BipD-mediated mitophagy induced by *B. pseudomallei* infection

In order to detect whether mitophagy could be induced by CUL3-based IMMT ubiquitination after *B. pseudomallei* infection, we generated *Cul3* and *Immt* knockout RAW264.7 cells, respectively. The levels of mtDNA were significantly declined by infection with *B. pseudomallei* WT, while these changes were partially abolished under Δ*bipD* strain infection in wild-type RAW264.7 cells treated with scrambled gRNAs. However, these levels remained unchanged in the absence of either CUL3 or IMMT (Fig. 7a, c). Concurrently, analogous observations were made in the levels of TIM23 and HSP60 (Fig. 7b, d). As delineated in Fig. 7e, f, in *B. pseudomallei* WT-infected cells, a clear BipD-dependent diminishment in the colocalization of mitochondria with LC3 was observed. However, this influence was entirely abrogated in both *Cul3*⁻/⁻ and *Immt*⁻/⁻ RAW264.7 cells. Consistently, loss of either CUL3 or IMMT reduced the colocalization of mitochondria with lysosome upon *B. pseudomallei* WT infection, while no obvious differences were found when infected with Δ*bipD* mutant (Fig. 7g, h). Besides, to further assess the requirement for IMMT K211 in mitophagy, mt-Keima assay was used to assess the mitophagy level induced by *B. pseudomallei* infection. As expected, *B. pseudomallei* infection induced a distinct mitophagy signal in IMMT-Flag overexpression cells (Fig. 7i). In contrast, this alteration was abolished by substitution of K211 of IMMT (Fig. 7i), indicating that the K211 of IMMT is required for *B. pseudomallei*-induced mitophagy occurrence.

Subsequently, we evaluated mtROS production and discerned that activation of wild-type RAW264.7 cells through *B. pseudomallei* Δ*bipD* infection culminated in markedly elevated mtROS levels, as compared to exposure to either *B. pseudomallei* WT or Δ*bipD:bipD* strains. These findings were congruent with prior results, and the effects were neutralized upon *Cul3* or *Immt* knockout (Fig. 7j). Similar influences were investigated concerning the regulation of the corresponding intracellular survival of *B. pseudomallei* (Fig. 7k). To corroborate whether IMMT ubiquitination, mediated by the KLHL9/KLHL13/CUL3 complex, impacts the virulence and intracellular viability of *B. pseudomallei* in vivo, BALB/c mice were subjected to infection with *B. pseudomallei* WT or Δ*bipD* via intraperitoneal injection, along with MLN4924 treatment. We visualized that more than half of *B. pseudomallei* WT-infected mice died within 2 days, whereas most (8 out of 12) mice survived *B. pseudomallei* WT challenge in the presence of MLN4924 (Fig. 7l). However, no significant effect of MLN4924 was assessed on *B. pseudomallei* Δ*bipD* infected mice (Fig. 7l). Consistently, bacterial loads in spleen and liver were obviously lower under MLN4924 treatment in response to *B. pseudomallei* WT infection, while no clear differences were visualized with *B. pseudomallei* Δ*bipD* infected (Fig. 7m). Collectively, these data demonstrate that BipD-mediated ubiquitination of IMMT depending on KLHL9/KLHL13/CUL3 complex is essential for mitophagy, assuming a crucial role in intracellular survival of *B. pseudomallei*.

## Discussion

Through IP-MS, this study unveiled that *B. pseudomallei* BipD, functioning as a mitophagy adapter, interacts with the KLHL9/KLHL13/CUL3 complex to act as an E3 ubiquitin ligase. Subsequently, combined with ubiquitinome MS screening, we identified a mitochondrial substrate IMMT, which is ubiquitylated by BipD-mediated modification, was essential for induction of host cell mitophagy responsive to the infection of *B. pseudomallei*. Furthermore, BipD cooperated with KLHL9/KLHL13/CUL3 complex to promote the K63-linked ubiquitination of IMMT at the site K211, and then autophagic degradation. And CUL3 and IMMT are required for *B. pseudomallei*-induced mitophagy to clear mtROS for promoting its intracellular survival. This research provides profound insights into the role of mitophagy, potentially paving the way for the formulation of more effective vaccines and

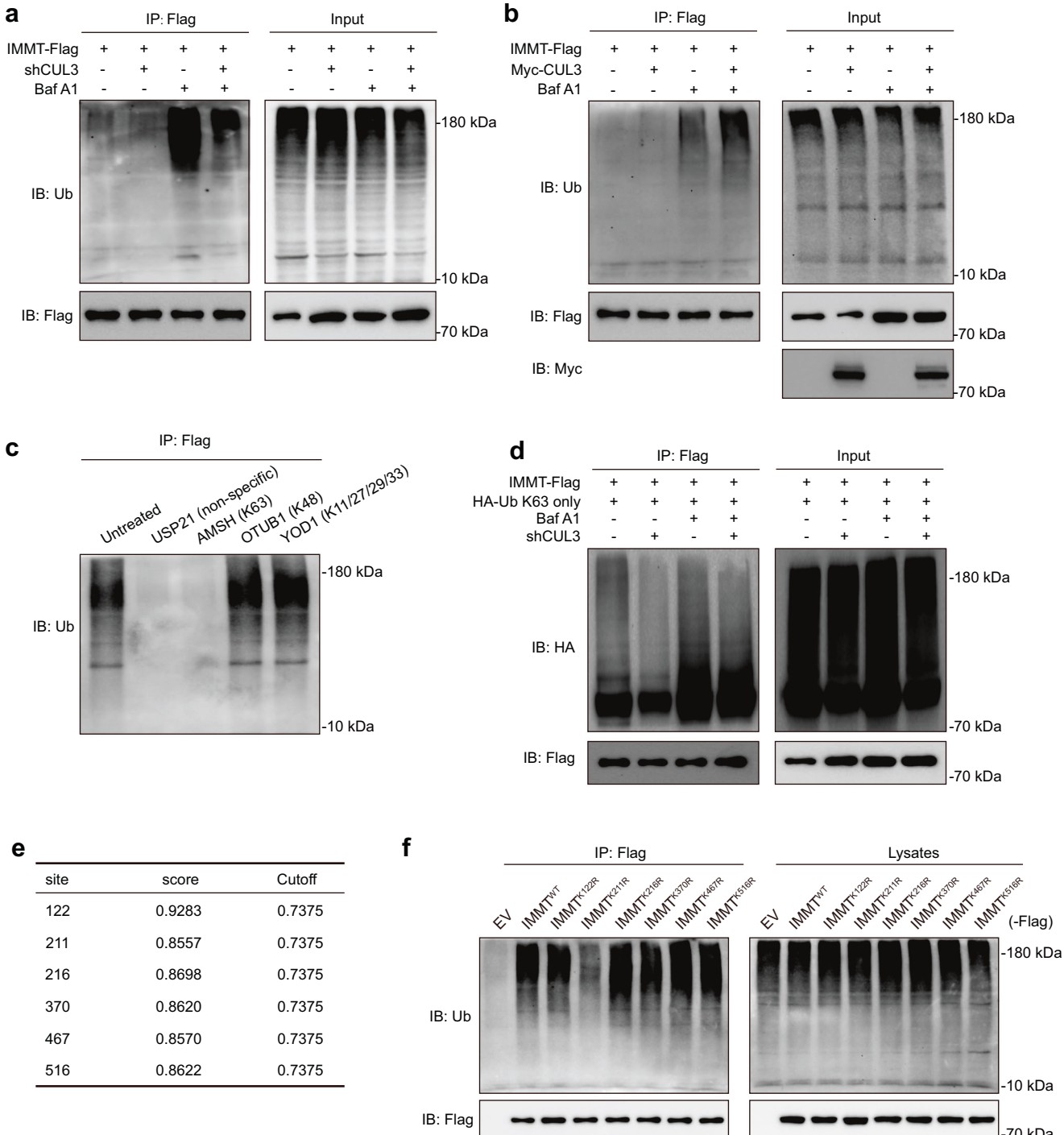

**Fig. 6 | CUL3 ubiquitinate IMMT with the K63-linked ubiquitin chains. a, b** Co-IP analysis determined the effect of CUL3 on IMMT ubiquitination in CUL3 knockdown or CUL3-overexpressing HEK293T cells. Cells were subjected to *B. pseudomallei* infection after Baf A1 pretreatment. **c** UbiCRest assays detected the ubiquitination type of IMMT in IMMT-Flag expressing HEK293T cells after *B. pseudomallei* infection using DUBs USP21, AMSH, OTUB1 and YOD1. **d** Co-IP analysis determined the effect of CUL3 on the K63-linked ubiquitin modification of IMMT in HEK293T cells. Cells were infected with *B. pseudomallei* after co-transfected with IMMT-Flag and HA-Ub K63 only plasmids in the presence of Baf A1. **e** The potential ubiquitination sites on IMMT predicted by GPS-Uber. **f** Co-IP analysis of ubiquitination sites on IMMT. Cells were transfected with IMMT mutant plasmids, collected IMMT mutants by IP and detected by western blot. Results characteristic of three separate trials. Source data were provided as a Source Data file.

therapeutic interventions targeted at reducing the substantial melioidosis mortality in developing countries.

Recently, several bacteria and viruses have reported to develop well-organized strategies to subvert mitophagy by PINK1/Parkin-independent manners. Both matrix protein of HPIV3 and glycoprotein of Hantaan virus (HTNV) induces mitophagy by translocating to mitochondria and then interacting with TUFM[4,5]. Furthermore, drawing from recent findings[8], it appears that distinct pathogens, such as *L. monocytogenes*, may invoke unique pathways for mitophagy induction. Specifically, NLRX1 has been recognized as a mitophagy receptor,

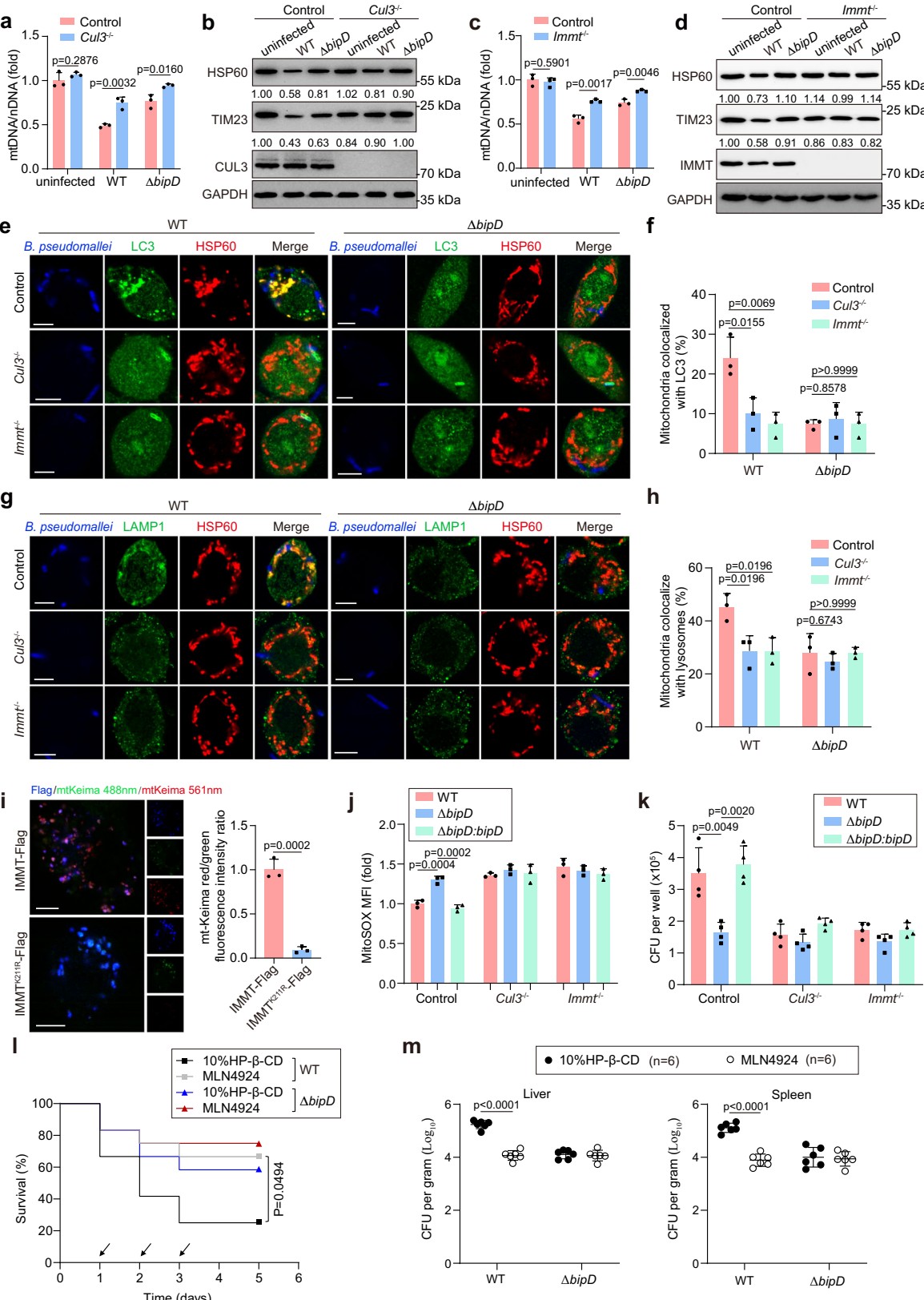

highlighting the intricate and diverse mechanisms through which pathogens may instigate mitophagy. In this study, although known primarily as a needle tip protein, BipD now has been clarified to enter the cytosol of infected cells by T3SS secretion to access its intracellular substrates to exert some of its functions. Of note, we found BipD could not induce mitophagy without mitochondria damaged stimuli,

suggesting the existence of bacterial proteins or toxins led to mitochondria damaged during *B. pseudomallei* infection, which will probably require an unbiased genetic screen assay for further research. In addition, given that Type III secretion is a tightly regulated process initiated only after host cell contact[36], despite *bipD* mutant was observed to hypersecrete the known T3SS effectors into the bacterial

**Fig. 7 | CUL3 and IMMT are important for function of BipD in mitophagy induction upon *B. pseudomallei* infection. a, c** Effect of CUL3 and IMMT on *B. pseudomallei* infection induced mitophagy. **b, d** Western blot analysis of mitochondrial markers in *B. pseudomallei*-infected *Cul3*[−/−] and *Immt*[−/−] RAW264.7 cells, respectively. **e, g** Colocalization of HSP60 with LC3 or LAMP1 was visualized by confocal microscopy in *Cul3*[−/−] or *Immt*[−/−] RAW264.7 cells, respectively. Cells were infected with *B. pseudomallei* WT or Δ*bipD*. Scale bar, 5 μm. **f, h** Quantification of cells with the mitochondria colocalized with LC3 or lysosomes in (**e**) and (**g**), respectively. 50 cells were calculated in all groups from three distinct trials. **i** Representative confocal images of *Immt*[−/−] RAW264.7 cells expressing IMMT-Flag or IMMT[K211R]-Flag. Mt-Keima assay was used to assess the mitophagy level induced by *B. pseudomallei* infection. Scale bar, 5 μm. **j** MitoSOX staining and flow cytometry analysis detected the impact of CUL3 or IMMT on the mtROS production after *B.* *pseudomallei* WT, Δ*bipD* or Δ*bipD:bipD* infection in RAW264.7 cells. **k** Detection of the effect of CUL3 or IMMT on intracellular survival of *B. pseudomallei* in RAW264.7 cells. Cells were under treatment in (**i**). **l** Survival ratio of *B. pseudomallei* WT or Δ*bipD* infected mice in the presence of 10% HP-β-CD or MLN4924 treatment. Mice models were infected with *B. pseudomallei* WT or Δ*bipD*, and treated with 10% HP-β-CD or MLN4924 every day for the first three days after infection (arrow point, *n* = 12 mice per group), and then continued to observe until day 5. **m** *B. pseudomallei* loads in liver and spleen from the mice treated as in (**l**) (*n* = 6 mice per group). The above data were from three or more independent experiments and showed means ± SD. Two-tailed Student's *t*-test (**a, c, i, m**), One-way ANOVA followed by Tukey post hoc test (**f, h, j, k**) and Mantel-Cox log-rank test (**l**) were used for data analysis. *n* = 3 per group (**a, c, f, h–j**), *n* = 4 per group (**k**). Source data were provided as a Source Data file.

culture supernatant[37], it is still not certain that what effect *bipD* deletion may have on the secretion of effectors and translocators when *B. pseudomallei*-infected host cells. Particularly, we have shown here that BopA is unlikely required for mitophagy initiation in response to *B. pseudomallei* infection, although it is proved to inhibit the LAP process[28,29]. Therefore, we reason that BopA and BipD are two important molecules, which are essential for *B. pseudomallei* to escape from the immune clearance of host defense response through different mechanisms.

The BTB superfamily is a multifaceted assembly comprising the KLHL, KBTBD, and KLHDC subfamilies. At present, at least 42 KLHL genes have been defined[31]. Especially, KLHL9 and KLHL13 are the most recent divergences among KLHL family members. While the KLHL9/KLHL13/CUL3 complex has been primarily implicated in regulating cell cycles, insulin resistance and cancer[38,39], its role in infectious diseases has remained largely unexplored. Generally, the KLHL proteins are characterized by their conserved BTB, Back and Kelch domains. In our study, BipD was found to interact with KLHL9 and KLHL13 through binding with both the Back and Kelch domains, despite prior studies have assigned few functions to the Back domain. The involvement of KLHL9 and KLHL13 in the ubiquitination process has been substantiated, as they are known to bind to CUL3, belonging to the NEDD8 family of RING E3 ligases[40]. Here, BipD recruits KLHL9/KLHL13/CUL3 complex to function as E3 ligase, resulting in ubiquitylating the damaged mitochondria after *B. pseudomallei* infection. Increasing evidence illuminates that pathogenic bacteria have developed the means to interfere extensively with different stages of ubiquitination and deubiquitination pathways for their strengths[41–43]. Noteworthy examples include *Salmonella* SopA and enterohemorrhagic *E. coli* NleL, which mimic eukaryotic HECT E3 ligases[44,45], while *Salmonella* SseL functions as a DUB in vivo[46,47]. *B. pseudomallei* TssM, the homologous protein of *B. mallei* TssM, acts as a DUB, and has been currently identified to inhibit the activation of NF-kB by promoting its substrates TRAF-3, TRAF-6 and IκBα deubiquitination[48]. In addition, CHBP (Cif homolog from *B. pseudomallei*) can catalyze the deamidation of Gln[40] in ubiquitin and NEDD8 to inhibit Cullin RING E3 ubiquitin ligases (CRL) activity in vitro[49], and further be verified to activate MAPK/ERK signaling in cellular assays which is independent of CRL inhibition[50,51]. Of note, many pathogenic *B. pseudomallei* strains lack the *Cif* gene, suggesting that CHBP is not an essential virulence factor in *B. pseudomallei*[51]. In summary, according to the above discoveries and our current results, it is demonstrated that *B. pseudomallei* has evolved a large array of mechanisms to survive and replicate within eukaryotic cells, which still remains to be further defined.

Currently, only a few substrates of the RING E3 ligase KLHL9/KLHL13/CUL3 complex have been identified. Aurora B has been shown to directly bind to the substrate-recognition domain of KLHL9 and KLHL13 and is ubiquitylated dependent on CUL3[38]. Analogously, the insulin receptor substrate-1 has been implicated in regulating insulin resistance via proteasomal degradation mediated by KLHL9/KLHL13/ CUL3[39]. The identification of the endogenous substrate(s) of the KLHL9/KLHL13/CUL3 complex remains a complex and challenging pursuit. Remarkably, the majority of mitochondrial proteins characterized as mitophagy receptors are situated on the OMM. However, the probability of an inner mitochondrial membrane (IMM) protein serving as a signaling platform of mitophagy has not been excluded[34,52]. Here, we found that IMMT was identified as a substrate of KLHL9/KLHL13/CUL3 complex, which is an essential protein situated on IMMs and interacts with several OMM proteins and orchestrates mitochondrial architecture[53–55]. This work has revealed that the mitochondrial stress induced upon *B. pseudomallei* infection may precipitate a mitochondrial integrity loss and OMM severing, and in turn contributes to a possibility for IMMT exposure and ubiquitination, which is involved with Parkin-independent, DRP1 mediation. Previous evidence has indicated that DRP1 can be activated in different ways to drive mitochondria division in mammals[34,56]. It stands to reason that the role of DRP1 in this model of mitophagy may participate in providing mitochondrial fragments small enough to be cleared by autophagy[57], but the detailed mechanism needs to be further investigated. Additionally, a preceding study has demonstrated that the autophagy adapter SQSTM1 recruits two CRL subunits Keap1 and Rbx1 to the mitochondria and then promotes OMM ubiquitination[58]. While, *B. pseudomallei*-induced mitophagy is more likely to require SQSTM1 to interact with ubiquitinated IMMT, thereby recruiting LC3-bound autophagosomes to the mitochondria. Suggesting an explanation that IMMT could not function as a mitophagy receptor directly associated with LC3 in the absence of a predicted LIR motif.

Our findings identify BipD as an effector to hijack host E3 ligase KLHL9/KLHL13/CUL3 complex, adding a mechanism of bacterial subversion of host innate immune response. Remarkably, the control of IMMT degradation by a NEDD8 RING E3 ligase, involving ubiquitination modification mediated by the bacterial effector, represents a sophisticated mechanism. The evolution of such an efficient strategy by *B. pseudomallei* to hijack host mitophagy for evading destruction stands as a testament to the continuously evolving war between the host and the infecting microbe. It will be worthwhile to continue to look for other sophisticated pathways that pathogen effectors manipulate host immune response, since different mechanisms are likely to be found for other pathogens.

## Methods
### Ethic statements
All animal studies were approved by the Laboratory Animal Welfare and Ethics Committee of the Third Military Medical University (AMU-WEC20223354), the experimental design and experimental process meet the requirements of animal ethics and animal welfare.

### Bacterial strains
The wild-type and mutant strains of *B. pseudomallei* were derived from *B. pseudomallei* BPC006[59]. *E. coli* S17-λpir or DH5α strain was used as

plasmids donor strain or cloning host. Bacteria were grown in Luria–Bertani (LB) medium as appropriate. The experimental operations of *B. pseudomallei* were carried out under standard laboratory conditions (biosafety level 3, BSL-3).

## Cells

HEK293T, RAW264.7, peritoneal macrophage cells (PMs), homozygous *Cul3*−/− and *Immt*−/− RAW264.7 cells were cultured in DMEM medium (Thermo Fisher, C11995500BT) supplemented with 10% FBS (Gibco, 10091148) and 1% glutaMAX (Gibco, 35050079). *Cul3* or *Klhl9* or *Klhl13* stably knocked down RAW264.7 or HEK293T cells were maintained in DMEM supplemented with 10% FBS, glutaMAX and 2 µg/mL puromycin (Thermo Fisher, A1113803). All the above cells were cultured at 37 °C with 5% $CO_2$. Infection of the cells by *B. pseudomallei* was performed as described previously[60]. Briefly, cells were seeded overnight, with or without pretreatment using 10 µM CCCP (Sigma-Aldrich, C2759), 20 µM Mdivi-1 (Sigma-Aldrich, M0199), 0.5 mM MitoTEMPO (Sigma-Aldrich, SML0737), 300 nM MLN4924 (ApexBio, B1036), 10 µM MG132 (ApexBio, A2585) or 100 nM Baf A1 (ApexBio, A8627) for 2 h or 20 µM CHX (ApexBio, A8244) for the indicated time points according to requirement, and infected with mid-log culture of *B. pseudomallei* at a multiplicity of infection (MOI) of 10 or at the indicated MOI for 1 h. Then cells were washed twice with prewarmed PBS and incubated in fresh medium containing 250 µg/ml kanamycin (TIANGEN, RT503) to kill the extracellular bacteria. At indicated time points, infected cells were washed three times with PBS and collected for subsequent experiments.

## Infection of mice

Specified Pathogen Free (SPF) sex-matched BALB/c mice at approximately 6–8 weeks (Vital River Laboratory Animal Technology Co., Ltd) were maintained under barrier conditions in a BSL-3 biohazard animal room at 25–27 °C and provided with free water and diet and a 12 h light/dark cycle. After one week of adaptive feeding, mice were injected intraperitoneal (i.p.) with $4 \times 10^5$ CFU of *B. pseudomallei* at day 1 and then injected i.p. with DMSO (Sigma-Aldrich, D4540), CCCP (5 mg/kg body weight), Mdivi-1 (50 mg/kg body weight) or MitoTEMPO (100 mg/kg body weight) at day 2, and evaluated the effect of mitophagy on bacterial load at day 3. Similarly, mice were treated i.p. with 10% HP-β-CD (Sigma-Aldrich, 332607) or MLN4924 (30 mg/kg body weight) every day for the first three days after infection, and then continued to observe until day 5. Liver and spleen were harvested and lysed with 0.1% Triton X-100 (AMRESCO). Diluted lysates were plated on luria broth agar plates. Colonies were counted after 36 h of incubation at 37 °C. All animal experiments were approved by the Laboratory Animal Welfare and Ethics Committee of the Third Military Medical University (AMUWEC20223354).

## Antibodies

Antibodies targeted the following proteins were used in this study: HSP60 (Mouse Monoclonal Antibody, Beyotime, AF0186, western blot 1:1000, immunofluorescence 1:100), HSP60 (Rabbit Monoclonal Antibody, Beyotime, AF1771, western blot 1:1000), GAPDH (CST, 5174, western blot 1:1000), TIM23 (clone 32, BD Biosciences, 611222, western blot 1:1000, immunofluorescence 1:100), LC3A/LC3B (Thermo Fisher Scientific, PA1-16931, western blot 1:1000, immunofluorescence 1:300), Flag (Zenbio, R24091, western blot 1:1000), SQSTM1 (Invitrogen, MA5-27800, immunofluorescence 1:100), TOMM20 (Abcam, ab186735, immunofluorescence 1:250), GST (Abclonal, AE077, western blot 1:1000), KLHL9 (Abcam, ab230542, western blot 1:1000), KLHL13 (Thermo Fisher Scientific, PA5-31658, western blot 1:1000), Xpress (Thermo Fisher Scientific, R910-25, western blot 1:1000), Ubiquitinylated proteins (clone FK2, Millipore, 04-263, immunofluorescence 1:200), Cullin 3 (Abcam, ab75851, western blot 1:1000), Ubiquitin antibody (clone P4D1, Santa Cruz Biotechnology, sc-8017, western blot

1:200), DnaK (Abcam, ab69617, western blot 1:1000), Mitofilin (Abcam, ab137057, western blot 1:1000), Myc (Beyotime, AF0033, western blot 1:1000), HA (Thermo Fisher Scientific, 26183, western blot 1:1000), LAMP1 (Thermo Fisher Scientific, MA1-164, immunofluorescence 1:50). KIF5B (Abcam, ab167429, western blot 1:1000), DYNLL1 (Abcam, ab51603, western blot 1:1000), ATG5 (CST, 12994T, western blot 1:1000), ATG7 (CST, 8558T, western blot 1:1000), FIP200 (Abclonal, A14685, western blot 1:1000), PINK1 (CST, 6946, western blot 1:1000), Parkin (Thermo Fisher Scientific, PA513399, western blot 1:1000), NLRX1 (ABclonal, A4976, western blot 1:1000), BNIP3L (Zenbio, 381891, western blot 1:1000), BNIP3 (Zenbio, R23308, western blot 1:1000), FUNDC1 (Zenbio, 251933, western blot 1:1000), BAX (Zenbio, R22708, western blot 1:1000), BAK (Abclonal, A0204, western blot 1:1000), DRP1 (Abclonal, A2586,western blot 1:1000), Donkey anti-Mouse IgG (H+L) Highly Cross-Adsorbed Secondary Antibody, Alexa Fluor™ 594 (A-21203) and Goat anti-Rabbit IgG (H+L) Cross-Adsorbed Secondary Antibody, Alexa Fluor™ 568 (A-11011) were purchased from Thermo Fisher Scientific (immunofluorescence 1:1000). Anti-rabbit IgG, HRP-linked Antibody (7074) and anti-mouse IgG, HRP-linked Antibody (7076) were purchased from CST (immunofluorescence 1:5000).

## Plasmids construction

Primer sequences were showed in the Supplementary Data 4. The plasmids of pcDNA4-IMMT-Flag, pcDNA4-TIMM50-Flag, pcDNA4-Flag-BipD, pcDNA4-Flag-SopD2, pGEX-GST and pGEX-GST-BipD were synthesized from Sangon Biotech (Shanghai, China). The sequences encoding *Klhl9* and *Klhl13* were amplified by PCR using primers in the Supplementary Data 4. PCR fragments were ligated into pcDNA4-Flag plasmid using In-Fusion PCR cloning kit according to the manufacturer's instruction (Clontech, USA). DNA sequences encoding shRNA of *Klhl9*, *Klhl13* or *Cul3* were synthesized from the Beijing Genomics Institute (BGI) and digested by *Age* I (NEB, R3552) and *Eco*R I (NEB, R3101V), and cloned into pLKO.1 lentiviral plasmid using T4 DNA ligase (NEB, M0202S). DNA sequences encoding lysine site mutant of *Immt* gene were amplified from pcDNA4-IMMT-Flag by PCR using the primers in Supplementary Data 4, the PCR fragments were then digested by *Dpn* I (NEB, R0176S). The plasmids were transformed into DH5α competent cells (Sangon, B528413-0010) for production and purification.

## Western blot

Cells were washed twice with cold PBS and lysed in cold RIPA lysis buffer (Beyotime, P0013B) containing EDTA-free protease inhibitor cocktail (Roche, 04693159001). The cell lysates were sonicated on ice until they became clear. The supernatants of the cell lysates were then isolated at $12,000 \times g$ for 10 min at 4 °C. Protein concentration was determined by BCA Protein Assay (Thermo Fisher, 23227) and equal amounts of proteins were denatured at 100 °C for 5 min in 1× Loading buffer (10% glycerol, 50 mM Tris-HCl (pH 6.8), 2% β-Mercaptoethanol, 0.02% Bromophenol blue, 2% SDS). Denatured proteins were separated by SDS-PAGE and then transferred to polyvinylidene difluoride (PVDF) membrane (Millipore, IPVH00010). After incubation with primary and horseradish peroxidase (HRP)-conjugated secondary antibodies, protein bands were detected by SuperSignal ECL Chemiluminescent Substrate Detection Kit (Invitrogen, 34580). Blots were scanned with the ChemiDoc Touch System (Bio-Rad USA) and analyzed using Image Lab™ Software (version 6.0.0, build 25, Bio-Rad USA) for Windows. GAPDH was used as loading control.

## Real-time qPCR (RT-qPCR) and measurement of mtDNA

Total RNA of lentivirus-mediated gene knockdown RAW264.7 cells or RAW264.7 cells transfected with the indicated siRNAs were extracted with Trizol (Invitrogen, 15596026) according to the manufacturer's instructions. cDNA was synthesized from 1 µg extracted RNA by PrimeScript™ RT reagent Kit with gDNA Eraser (TaKaRa, RR047A). The

cDNA samples were then amplified using Platinum™ SYBR™ Green qPCR SuperMix (Thermo Fisher, 11733046) on a CFX96 Real-Time qPCR system (Bio-Rad). Relative-fold expression for target genes was calculated by the $2^{-\Delta\Delta Ct}$ method relative to GAPDH.

Total DNA of cells induced mitophagy with CCCP or *B. pseudomallei* infection was extracted with TaKaRa MiniBEST Universal Genomic DNA Extraction Kit Ver.5.0 (TaKaRa, 9765) according to the manufacturer's protocols, and 10 ng DNA was detected by RT-qPCR described above. All primer sequences used in the study were listed in Supplementary Data 4. The ratio of mtDNA to nDNA was used as an estimate for the mtDNA copy number.

## Immunofluorescence
Cells were seeded on coverslips (NEST, 801011) in 24-well plates and transfected with the indicated plasmids. After incubation for 24 h, cells were washed twice with PBS and fixed in 4% paraformaldehyde (Beyotime, P0099) for 10 min, subsequently, cells were permeabilized with 0.3% Triton X-100 (amresco, 0694-1L), and blocked with 1% BSA at room temperature. Primary antibodies were then applied overnight at 4 °C. After washing three times with PBS, the fluorophore-conjugated secondary antibodies were incubated for 2 h at room temperature. Then, the nucleus was stained with DAPI (Thermo Fisher, 62248) and the coverslips were mounted onto glass slides using anti-fade mounting medium (Invitrogen, S36967). Confocal images were taken with the Leica SP8 confocal microscope (Leica Microsystems) and analyzed by the Leica Application Suite Las X (v2.0.1.14392) software. Specially, Images were acquired with a Zeiss LSM 900 confocal microscope equipped with an Airyscan superresolution imaging module. Methods of images acquisition were followed by the description of Oshima et al.[34]. Cells were viewed by Airyscan image processing.

## Generation of stable knockdown cell lines
To generate RAW264.7 or HEK293T cells stably knockdown *Klhl9*, *Klhl13* or *Cul3*, the lentiviral expression pLKO.1-TRC cloning vector encoding shRNA for *Klhl9*, *Klhl13* or *Cul3* were co-transfected together with the lentiviral packaging plasmids psPAX2 and pMD2.G in HEK293T cells using lipofectamine 3000 (Invitrogen, L3000015) respectively according to the manufacturer's instructions. The culture medium was replaced after 24 h of transfection and collected at 48 h and 72 h. The collected medium was centrifuged and filtered using 0.22 μm filter. Then RAW264.7 or HEK293T cells were transduced with enveloped lentivirus using half-volume infection method and incubated with 8 μg/mL polybrene (Yeasen, 40804ES76). The culture medium was replaced every 24 h and the culture medium containing 2 μg/mL puromycin was applied 48 h later. The stably knockdown cells were generated after puromycin treatment for 72 h[4]. Protein and RNA were extracted to detect the efficiency of gene knockdown using western blot and RT-qPCR.

## Knockout of *Cul3* or *Immt* in RAW264.7 cells
Deletion of *Cul3* or *Immt* in RAW264.7 cell lines was accomplished by Cyagen (Chain). Briefly, gRNA targeting the mouse *Cul3* or *Immt* exons was designed and constructed according to *Cul3* and *Immt* sequences obtained from NCBI. The synthetic gRNA was incubated with Cas9 protein and transfected into RAW264.7 cells using electroporation. Single-cell were plated in 96-well plates and cultured in the prepared DMEM medium supplement with 10% FBS. When the confluence of cell clones generally reached 60%, the backup plating was performed. Extracted the DNA and amplified by PCR using the primers upstream and downstream of the gRNA target site and sequenced the amplified products to select the homozygous knockout clones. Cultivated the homozygous knockout clones and extracted protein to validation of knockout at protein level. Homozygous *Immt*$^{-/-}$ and homozygous *Cul3*$^{-/-}$ RAW264.7 cell were obtained and cultured in DMEM with 10% FBS and 1% glutaMAX at 37 °C in 5% $CO_2$.

## Electron microscopy
RAW264.7 or PMs cells were seeded at approximately 80% confluence in advance, then infected with *B. pseudomallei*. After 4 h of infection, cells were washed with PBS, digested with trypsin and fixed overnight in 2.5% glutaraldehyde in 0.1 M phosphate buffer (pH 7.2) at 4 °C. Thereafter, cells were fixed in 1% aqueous osmium, dehydrated with increasing concentrations of ethanol (30%, 50%, 70%, 80%, 90% and 100%), embedded, and sectioned. Ultrathin sections (40−60 nm) were stained with 2% aqueous uranyl acetate and lead citrate. The sections were observed under a transmission electron microscope (EM420) at 60 kV.

## Coimmunoprecipitation
HEK293T cells were seeded in 10 cm dishes and kept at 37 °C with 5% $CO_2$ in an incubator for 24 h. Then, cells were transfected at 60 − 80% density using Lipofectamine™ 3000 24 h before cell lysis according to the manufacturer's protocols. In brief, 0.5 ml IP lysis buffer (50 mM Tris, 150 mM NaCl, 0.1% Triton X-100, pH 7.5) containing protease inhibitor cocktail (Roche) per dish, protein was extracted for 10 min at 4 °C. After $13,000 \times g$ 4 °C centrifugation, the supernatant was taken for coimmunoprecipitation. 50 μl of the supernatant was taken for input, while the remaining was used for IP. The anti-Flag M2 Affinity Gel (Sigma-Aldrich, A2220) were washed 3 times using IP lysis buffer, and mixed with the above supernatant rotating 6 h at 4 °C. Unbound proteins were cleared by 2 min 4 °C centrifugation at $1500 \times g$, and used lysis buffer washing 3 times. Then, 100 μl Flag-peptide competition elution was performed by incubation with beads at 4 °C for 2 h. After centrifugation ($1500 \times g$ for 1 min), samples were either used for western blot analysis or stored at −80 °C.

## Immunoprecipitation-mass spectrometry (IP-MS) analysis
To determine the protein from HEK293T cells binding to BipD, we performed the IP-MS analysis. HEK293T cells in a 10 cm dish were transfected with 15 μg plasmid of Flag-BipD or Flag-SopD2 ($n = 4$ in each group). At 24 h post-transfection, cells were collected and lysed in lysis buffer (50 mM Tris-HCl, pH 7.4, 150 mM NaCl, 1% Triton X-100 and protease inhibitor cocktails) on ice for 30 min. Cell lysates were cleared by centrifugation at $14,000 \times g$ for 10 min at 4 °C and the supernatant were incubated with anti-Flag M2 resins for 6 h at 4 °C. The resins were washed three times with lysis buffer followed by eluted with 3× Flag peptide for 2 h at 4 °C. The IP samples were separated by SDS-PAGE and the stained gel was excised into 3 slices per sample, which were further subjected to in-gel digestion[61]. Briefly, the bands were reduced and alkylated using dithiothreitol and iodoacetamide, respectively. The protein samples were digested overnight with trypsin. The resulting peptides were dried in a SpeedVac vacuum concentrator and then resuspended in HPLC-grade water for further MS/MS analyses.

## GST pulldown
The gene encoding *B. pseudomallei bipD* was cloned into the pGEX-6p-2 plasmid and expressed in *E. coli* BL21 (DE3) cells (TIANGEN, CB105-02) and purified with Glutathione Sepharose (GE Healthcare). cDNA encoding human His-KLHL9 or His-KLHL13 was cloned into the pcDNA4-HisMax plasmid and transfected into HEK293T cells. Then, cells lysates were collected in Pull-Down Lysis Buffer and centrifuged at $13,000 \times g$ for 15 min. GST pulldown assay was carried out with an Pierce™ GST Protein Interaction Pull-Down Kit (Thermo Scientific, 21516) according to the manufacturer's protocols.

## Ubiquitin proteomics analysis by LC-MS/MS
To verify the substrate protein of BipD to induce mitophagy, we performed the ubiquitin proteomics analysis. pET28a-His6-TUBE plasmid was transformed into *E. coli* BL21 (DE3) cells and used IPTG to induce protein expression. Then purified His6-TUBE using Ni-NTA-Sepharose

(GE Healthcare) chromatography. Purified proteins were covalently immobilized to NHS-activated Sefinose (Sangon Biotech, C600024) according to the manufacturer's instructions. HEK293T cells in 15-cm dish were transfected with Flag-BipD or Flag-EV plasmids for 16 h and then treated with 20 mM CCCP was added to induced mitochondrial damage for 8 h ($n = 3$ in each group). Cells lysates were harvested and incubated with TUBE-conjugated beads to enrich ubiquitinated proteins. Ubiquitinated proteins were eluted and dried in SpeedVac vacuum concentrator (Thermo Scientific). The dried samples were resuspended in 8 M urea, PBS pH 8.0, followed by in-solution digestion. The resulting peptide was desalted on C18 SepPak solid-phase extraction cartridges (Welch Materials, 00559-11002) and underwent off-line high pH reverse phase LC separation. 72 fractions were concatenated back to 12 fractions in 6 cycles and dried in a vacuum concentrator, and the dried samples were stored at 80 °C for further LC-MS/MS analysis.

The resulting peptides were dissolved in HPLC-grade water prior to LC–MS/MS analyses. A hybrid ion trap orbitrap mass spectrometer (LTQ Orbitrap Velos, Thermo Scientific, Waltham, MA, USA) was applied for peptide analyses. Homemade C18 analytical columns (75 μm × 150 mm) were equipped with 4 μm of 100 Å Magic C18AQ silica-based particles (Michrom BioResources Inc., Auburn, CA, USA). Eluted peptides were electrosprayed directly into the mass spectrometer for MS and MS/MS analyses in data-dependent acquisition mode.

The raw files were processed with Mascot Daemon (version 2.3.02, Matrix Science). Tandem mass spectra were searched against nonredundant human protein database downloaded from the UniProt website (www.uniprot.org). The precursor mass tolerance and the fragment mass tolerance were set at 20 ppm and 0.8 Da, respectively. Meanwhile, carbamidomethylation of cysteine residues was set as a fixed modification, and oxidation of methionine was set as a variable modification. Trypsin was set as a digestion enzyme with a maximum of two missed cleavages. The false discovery rates (FDRs) of peptides and proteins were restricted to under 1%.

## Flow cytometry

To detect the mtROS in RAW264.7 cells or mouse PMs after *B. pseudomallei* infection, cells were infected with *B. pseudomallei* at the indicated MOI for indicated time points. Cells were washed with PBS for three times, stained by 5 μM mitoSOX (Thermo Fisher, M36008) for 15 min at 37 °C in dark place, washed with PBS for three times, and then collected with PBS in the sample tubes of flow cytometry according to the manufacturer's instructions. To detect the MMP in RAW264.7 cells or mouse PMs, cells were infected with *B. pseudomallei* as described above. Cells were washed with PBS for three times, stained by 100 nM TMRM (Invitrogen, T668) for 30 min at 37 °C in dark place, washed with PBS for three times, and then collected with PBS in the sample tubes of flow cytometry according to the manufacturer's instructions. The MFI data were acquired by the Fortessa flow cytometer (BD Biosciences) and analyzed by FlowJo software.

## UbiCRest assays

IMMT-Flag was purified from HEK293T cells transfected with IMMT-Flag plasmid using anti-Flag M2 affinity gel and resuspended in dilution buffer (25 mM Tris pH 7.5, 150 mM NaCl). DUBs used in the UbiCRest assays were purchased, including USP21 (Lifesensor, DB-0509-0025), YOD1 (Fitzgerald, 80R-2723), AMSH (Lifesensor, DB-0301-0025) and OTUB1 (Lifesensor, DB-0201-0025), and resuspended in dilution buffer with 10 mM DTT at 2× working concentration (5 μM USP21, 5 μM AMSH, 5 μM OTUB1, 5 μM YOD1). Preincubated DUBs at room temperature for 10–15 min, then mixed 10 μl of ubiquitinated substrates IMMT-Flag with 10 μl of DUB solution and incubated the reaction mixture at 37 °C according to the protocols[35]. Then mixed them with 4× LD sample buffer to stop the reaction immediately and detected by western blot analysis for ubiquitin using ubiquitin antibody.

## The TEM secretion assay

T3SS-mediated injection of BipD was evaluated by the FRET-based substrate CCF2-AM (Thermo Fisher Scientific, K1023). RAW264.7 cells were seeded onto φ20mm confocal dish (NEST, 801001) in DMEM with 10% FBS at $1.5 \times 10^5$ cells/well. The gene encoding a β-lactamase was cloned into the vector of pUCP28T with the gene of *bipD* and *bopA* to generate BipD-TEM and BopA-TEM fusion proteins. RAW264.7 cells were then infected with the *B. pseudomallei* which expressed BipD-TEM or BopA-TEM by the fusion expression plasmid using an MOI of 10. After 1 h of infection the cells were washed twice with PBS and incubated in fresh medium containing 250 μg/ml kanamycin to kill the extracellular bacteria. At 4 h, cells were washed twice with PBS, followed by incubation of infected cells with the fluorescent substrate CCF2-AM at 2 μg/mL according to the manufacturer's instructions. Translocation of BipD-TEM or BopA-TEM fusion protein into CCF2-loaded cells was determined after incubating at room temperature and avoiding light for 1 h, by counting the number of blue fluorescent cells in images taken with the Leica SP8 confocal microscope (Leica Microsystems), with excitation light of 405 nm and absorption light of 447 and 520 nm dual filters to collect absorption peaks. The loaded cells that did not exhibit TEM fusion secretion showed green color.

## Construction of *bipD* deleted strain (Δ*bipD*) and its complemented strain (Δ*bipD:bipD*) of *B. pseudomallei*

*B. pseudomallei* (BPC006, NC_018529.1 and NC_018527.1) was selected as the parent strain for the construction of *bipD* deletion mutants. *B. pseudomallei bipD* mutant strains were generated by double-crossover allelic exchange using the λpir-dependent vector pK18mobsacB which contains the *sacB* gene for counter-selection. Briefly, approximately 1 kb fragments upstream and downstream of *bipD* gene (GenBank accession no. BPC006_RS26315) was amplified from BPC006 genomic DNA using the primers listed in Supplementary Data 4. PCR amplifications were carried out with the SimpliAmp™ PCR cycler (Applied Biosystem, USA) using the following conditions: 95 °C for 5 min, 35 cycles each of 10 s at 95 °C, 30 s at 58 °C, 1 min at 72 °C, and a final extension of 5 min at 72 °C. PCR products were cloned into pK18mobsacB vector using In-Fusion PCR cloning kit (Clontech, USA) according to manufacturer's instruction. The construct was subsequently conjugated into *B. pseudomallei* through *E. coli* S17-λpir. Clones with successful single crossover by homologous recombination were first selected on LB with 250 μg/ml kanamycin and 50 μg/ml gentamycin. Positive clones were then grown in LB containing 15% sucrose with 50 μg/ml gentamycin at 24 °C for 2–3 days to counter against the sacB present in pK18mobsacB. The mutant of *bipD* deletion which were the successful double crossover clones were screened by colony PCR.

The full-length of *bipD* gene were amplified from the genomic DNA of *B. pseudomallei* by PCR using appropriate primers listed in Supplementary Data 4. PCR product was cloned into broad-host-range vector pUCP28T by T4 DNA Ligase (NEB, M0202S) to generate pUCP28T-BipD. Then, the recombinant plasmid pUCP28T-BipD was electroporated into *B. pseudomallei*. Expression of BipD was confirmed by immunoblot.

## Mt-Keima mitophagy detection analysis

To determine whether *B. pseudomallei* infection induced mitophagy, RAW264.7 cells were incubated with adenovirus expressing mt-Keima (HANBIO) at a MOI of 10, and facilitated infection using polybrene according to the protocols. At 24 h post-infection, cells were treated with 10 μM CCCP or infected with *B. pseudomallei* at a MOI of 10 for 4 h. Then, cells were washed with PBS for 3 times and observed at 488 nm (green) or 561 nm (red) light. The mitophagy ratio could measure by the red mean fluorescence intensity (MFI)/green MFI. Data were analyzed by FlowJo software.

## Quantification and statistical analysis

Statistical analysis was performed using GraphPad Prism (v8.0) and ImageJ 1.53e software. Two-tailed Student's *t* test was used for two-group comparisons. One-way ANOVA followed by Tukey post hoc test and two-way ANOVA followed by Tukey post hoc test were used for multiple comparisons. Log-rank (Mantel-Cox) test was used to analyzed the mouse survival curves and statistical differences. Significance was indicated in the figures and figure legends respectively. All assays data were performance at least three biological replicates and expressed as the means ± SD. The statistical tests and the numbers of samples were indicated in the corresponding figure legends.

## Reporting summary

Further information on research design is available in the Nature Portfolio Reporting Summary linked to this article.

## Data availability

The proteomics data used in this study are available in the ProteomeXchange database at PXD051631. Source data are provided with this paper.

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

## Acknowledgements

We thank Yunn-Hwen Gan for providing the pK18mobsacB and pUTC28T plasmids, Tamotsu Yoshimori for providing the GFP-LC3B plasmid, Xiaofeng Zhu for providing the pCS2-Myc-CUL3 plasmid, Wanyan Deng for technical assistance, and Haibo Wu for helpful suggestions. The rigorous work described herein was made possible by generous support from the National Natural Science Foundation of China (Nos. 32270190 and 31970135, Q.L., 82272350, X.M., 32300161, Z.H.) and the Army Medical University Youth Development Foundation (2020XQN06, Q.L.).

## Author contributions

Conceptualization, Q.L., Q.Z., X.M, X.L., D.N., and C.R.; Investigation, D.N., C.R., Z.T., W.Y., P.W., J.C., Y.X., W.L., Z.Z., and Z.H.; Data analysis, D.N., C.R., Z.T., J.Y., H.C., Y.L. and Z.Z.; Supervision, Q.L., Q.Z., X.M., and X.L.; Writing-Original Draft, Q.L., D.N., C.R., Z.T., and P.W.; Funding Acquisition, Q.L., Q.Z., and X.M. All authors read and approved the final manuscript.

## Competing interests

The authors declare no competing interests.
