## [Peer Review File · Nature Communications]

Burkholderia pseudomallei BipD modulates host mitophagy to evade killingEditorial Note: This manuscript has been previously reviewed at another journal. This document only contains reviewer comments and rebuttal letters for versions considered at *Nature Communications*.

Reviewers Comments:

Reviewer #1 (Remarks to the Author):

Nan and colleagues explore host pathogen interactions in the context of Bukholderia infection manipulating mitophagy to its advantage. The concept of pathogens hijacking mitophagy for their benefit has been reported previously in several studies, and therefore the major advance of the present study is focused around the mechanisms by which Bukholderia manipulates mitophagy. The authors report a role for the type III secretion system needle tip protein BipD as an interactor of KLHL9 and KLHL13 in complex with CUL3 to ubiquitinate mitochondria and drive mitophagy. The mitochondrial inner membrane protein IMMT was identified as a ubiquitination target for the ligase complex at K211 which was proposed to be required for initiating mitophagy. Overall, the proposed mechanisms very quite interesting, but they are not convincingly supported by the data. Evidence for exposure of the mitochondrial inner membrane and its ubiquitination is lacking, as is direct evidence showing that IMMT K211 is the primary driver of mitophagy initiation. In addition, it is conceptually challenging to understand how widespread depolarization and presumably rupturing of mitochondrial membranes would be beneficial to the pathogen given that this would ultimately result in significant cytochrome c release and cell death via apoptosis. It is also very notable that none of the electron microscopy images of mitochondria in the manuscript undergoing mitophagy show any evidence of outer membrane rupture and inner membrane exposure as has been clearly shown by others, including during inner membrane dependent forms of mitophagy (e.g. Oshima et al (2021) J Cell Biol, McArthur et al (2018 Science), Yoshii et al (2011) J Biol Chem)). This manuscript would therefore benefit with more mechanistic insight to convincingly support the authors key findings of mitophagy, including analyses of inner membrane exposure, assessments for recruitment of the ligase complex as well as autophagy machineries to exposed inner membrane, and direct evidence of mitophagy initiation by IMMT. With stronger evidence supporting the authors conclusions, this manuscript would represent a very important and interesting discovery that would be of broad interest to a variety of researchers.

1. Figure 1: To confirm and strengthen the conclusion that the delivery of mitochondria to lysosomes using mtkeima during infection is mitophagy dependent can the authors conduct the infection experiment in cells defective for mitophagy (e.g. FIP200 knockout/knockdown)?
2. ED Fig 1-B: Can the authors undertake microscopy of infected cells with a mitochondrial marker unaffected by membrane potential loss (e.g. mitotracker green) together with TMRE staining to provide visual evidence of mitochondrial depolarization? This will also help to ascertain what proportion of the mitochondrial network is becoming depolarized.
3. Can the authors assess whether there is clear evidence of mitochondrial inner membrane exposure and ubiquitination following infection using microscopy? It is recommended the authors use the well established techniques utilized by others, Oshima et al (2021) J Cell Biol as an example, but also refer to the manuscripts cited in the general comments above. Typically, imaging of an outer membrane marker (Tom20) together with a matrix marker and ubiquitin staining are used to assess whether regions where the inner membrane is exposed (i.e. Tom20 negative, matrix marker positive) are ubiquitinated.
4. In addition to showing evidence of mitochondrial inner membrane exposure and ubiquitination, using microscopy, can the authors assess whether autophagy machineries are recruited (e.g. ATG13, WIPI2), including ubiquitin binding autophagy adaptors that are known to drive mitophagy (e.g. NDP52, Optineurin)?
5. Is BipD and the associated ubiquitin ligase complex recruited to the exposed inner membrane of mitochondria (as assessed by microscopy)?
6. Mdivi-1, originally developed as a Drp1 inhibitor, is a well established off-target compound and not a specific inhibitor of mitophagy (e.g. Bordt et al (2017) Dev Cell). Therefore, the experiments

using this compound as an autophagy inhibitor are not highly informative and the authors would be better served not including data using Mdivi-1.

7. To date, the factors known to result in rupture of the outer membrane during inner membrane mediated mitophagy are Parkin (Yoshii et al (2011) J Biol Chem), Bax and Bak (McArthur et al (2018 Science), and Drp1 (Oshima et al (2021) J Cell Biol). The authors have excluded Parkin, can they also assess whether Drp1 or BAX/Bak contribute to the infection induced inner membrane mitophagy they observe?

8. Can the authors utilise knockdown/knockout of IMMT and rescue with WT or K211A mutant IMMT followed by mtKeima mitophagy analysis to directly assess the requirement for IMMT K211 in mitophagy?

Reviewer #2 (Remarks to the Author):

This article uncovers something novel about the ability of *B. pseudomallei* to induce mitophagy to evade intracellular killing. It is well organized and presents several interesting pieces of data on the infection biology of the bacteria. However, there are several areas that require major clarifications and amendments.

1. The study did not address prior findings in the field on *B. pseudomallei*'s ability to induce autophagy and its interference in ubiquitination. Often, these studies are in contradiction to the current findings, and the authors could explain the discrepancies, eg due to differences in techniques and also the fact that the bacteria have several mechanisms in operation that are working in opposite directions. It depends on when all these effectors are expressed, and this point is often obscured when there is too high an MOI of infection or a careful kinetic study is not undertaken.

a) BopA has been shown to inhibit autophagy, and that has been shown to increase intracellular bacterial counts. However, the study did not examine whether the autophagy they observed in fact could be mitophagy instead. The authors should cite these papers PMID: 21412437, PMID: 18483470 and interact with these findings. Would a bopA mutant shows an opposite phenotype to bipD mutant in terms of mitophagy? Or the 2 are independent phenomena?

b) the authors also did not address previous study showing TssM PMID: 20335533 as a deubiquitinase. These factors are all in PMID: 29848489 bacteria. This could be addressed in the Discussion.

c) Cif is another factor of Bp, although it is not present in all strains of Bp. It can increase slightly intracellular counts and partly through inhibition of Cullin RING E3 ubiquitin ligase PMID: 29848489

2. Is the bipD mutant still able to secrete bsaT3SS effectors? Or that it is in fact causing over secretion of effectors? If the BipD mutation alters the secretion of effectors, including those that may have an opposite effect like BopA, then using the mutant to validate the infection and intracellular survival is problematic. Do the authors know the domain of BipD that interacts with the Cul3 complex? Through site directed mutagenesis, the authors could determine which mutations allow BipD to be translocated but it is now unable to interact. If this mutant sequence is complemented into the bipD mutant, and the ability to induce mitophagy and intracellular counts are measured, it will increase the validity of the claim.

3. BipD is also present in *B. thailandensis* but the authors claim that this doesn't happen in Bt infection. However, this claim may be wrong as Bt infection follows a slower kinetics. If the authors measure the effects with longer time points, they may see the same thing. If not, the authors could examine the protein sequence differences between BipD of the 2 species.

4. Figure 7. It helps if the authors could use densitometry to quantify their western blots where normalization to control proteins are shown. Figure 7d shows a different coloration on the strip on IMMT under IMMT- lanes. Please state either the 3 lanes were not done or to clarify.

RE: RE: *Burkholderia pseudomallei* BipD hijacks host KLHL9/KLHL13/CUL3 E3 ligase to ubiquitinate IMMT that initiates mitophagy to evade killing (NCOMMS-23-52856A)

Reply to the comments from the Board of Editors and Reviewers:

Reviewer Expertise:

Referee #1: mitophagy

Referee #2: *Burkholderia* pathogenesis, T3SS function

Reviewers Comments:

Reviewer #1 (Remarks to the Author):

Nan and colleagues explore host pathogen interactions in the context of *Burkholderia* infection manipulating mitophagy to its advantage. The concept of pathogens hijacking mitophagy for their benefit has been reported previously in several studies, and therefore the major advance of the present study is focused around the mechanisms by which *Burkholderia* manipulates mitophagy. The authors report a role for the type III secretion system needle tip protein BipD as an interactor of KLHL9 and KLHL13 in complex with CUL3 to ubiquitinate mitochondria and drive mitophagy. The mitochondrial inner membrane protein IMMT was identified as a ubiquitination target for the ligase complex at K211 which was proposed to be required for initiating mitophagy. Overall, the proposed mechanisms very quite interesting, but they are not convincingly supported by the data. Evidence for exposure of the mitochondrial inner membrane and its ubiquitination is lacking, as is direct evidence showing that IMMT K211 is the primary driver of mitophagy initiation. In addition, it is conceptually challenging to understand how widespread depolarization and presumably rupturing of mitochondrial membranes would be beneficial to the pathogen given that this would ultimately result in significant cytochrome c release and cell death via apoptosis. It is also very notable that none of the electron microscopy images of mitochondria in the manuscript undergoing mitophagy show any evidence of outer membrane rupture and inner membrane

exposure as has been clearly shown by others, including during inner membrane dependent forms of mitophagy (e.g. Oshima et al (2021) J Cell Biol, McArthur et al (2018 Science), Yoshii et al (2011) J Biol Chem)). This manuscript would therefore benefit with more mechanistic insight to convincingly support the authors key findings of mitophagy, including analyses of inner membrane exposure, assessments for recruitment of the ligase complex as well as autophagy machineries to exposed inner membrane, and direct evidence of mitophagy initiation by IMMT. With stronger evidence supporting the authors conclusions, this manuscript would represent a very important and interesting discovery that would be of broad interest to a variety of researchers.

1. Figure 1: To confirm and strengthen the conclusion that the delivery of mitochondria to lysosomes using mtkeima during infection is mitophagy dependent can the authors conduct the infection experiment in cells defective for mitophagy (e.g. FIP200 knockout/knockdown)?

Answer: We thank the reviewer's constructive suggestions. Canonical autophagy is essential for autophagosome formation and thus for degradation of select cargos like mitochondria¹. We next determined whether conventional autophagy is involved in *B. pseudomallei*-induced mitophagy. Both *B. pseudomallei*- and CCCP-induced decreases in the levels of mtDNA as well as mitochondrial proteins were blocked in ATG5 knockdown macrophages (Extended Data Fig. 2a-d in the revised manuscript). Furthermore, the number of red mt-Keima puncta was significantly reduced when ATG5 was knocked down, indicating a declined level of mitolysosomes (Extended Data Fig. 2e in the revised manuscript). Similarly, as shown in Extended Data Fig. 2f-h in the revised manuscript, the decrease in mtDNA induced by *B. pseudomallei* infection or CCCP treatment was also suppressed after siRNA-mediated knockdown of other autophagy associated genes (ATG7 and FIP200). Taken together, these results confirm that *B. pseudomallei* infection has the capability to induce mitophagy in mouse macrophages, which is dependent on the conventional autophagy. These results are now shown in Extended Data Fig. 2 and we point this out in the revised manuscript on page 5 line 102-110 Thanks.

2. ED Fig 1-B: Can the authors undertake microscopy of infected cells with a mitochondrial marker unaffected by membrane potential loss (e.g. mitotracker green) together with TMRE staining to provide visual evidence of mitochondrial depolarization? This will also help to ascertain what proportion of the mitochondrial network is becoming depolarized.

Answer: We thank the reviewer's constructive suggestions. We performed immunofluorescence staining with TMRE (100 nM) and Mitotracker Green (75 nM) which elucidated that *B. pseudomallei* infection resulted in a significant increase in the proportion of depolarized mitochondrial in RAW264.7 cells. The result is now shown in Extended Data Fig. 1c and we point this out in the revised manuscript on page 5 line 81-84 as "To address mechanisms of mitochondrial quality control upon *B. pseudomallei* infection, we measured the alteration of mitochondrial membrane potential (MMP, $\Delta\psi_m$) and found *B. pseudomallei* infection resulted in a significant loss of $\Delta\psi_m$ by using flow cytometric (Extended Data Fig. 1a,b) and microscopic analyses (Extended Data Fig. 1c)". Thanks.

3. Can the authors assess whether there is clear evidence of mitochondrial inner membrane exposure and ubiquitination following infection using microscopy? It is recommended the authors use the well established techniques utilized by others, Oshima et al (2021) J Cell Biol as an example, but also refer to the manuscripts cited in the general comments above. Typicly, imaging of an outer membrane marker (Tom20) together with a matrix marker and ubiquitin staining are used to assess whether regions where the inner membrane is exposed (i.e. Tom20 negative, matrix marker positive) are ubiquitinated.

Answer: We thank the reviewer's valuable suggestions. Firstly, we performed immunofluorescence staining with Ub (FK2 antibody, green), IMMT (red) and Tom20 (blue) in *B. pseudomallei*-infected RAW264.7 cells. Methods of images acquisition were followed by the description of Oshima et al ². As shown in the Extended Data Fig. 8d,e in the revised manuscript, we found a discontinuous fluorescence signal of TOMM20 and the colocalization of Ub and IMMT in response to *B. pseudomallei* infection. Furthermore, we repeated the TEM observation, and the TEM images revealed mitochondrial membranes were slightly severed after *B. pseudomallei* infection (Fig. 3g in the revised manuscript). Thus, these data indicate that OMM severing may contribute to a possibility for IMMT exposure and ubiquitination following infection. The result is now shown in Fig. 3g, Extended Data Fig. 8d,e and we point this out in the revised manuscript on page 8 line 178-180, and page 10 line 240-244, and discussed this observation detailly in the Discussion on page 15 line 407-422 as "This work has revealed that the mitochondrial stress induced upon *B. pseudomallei* infection may precipitate a mitochondrial integrity loss and OMM severing, and in turn contributes to a possibility for IMMT exposure and ubiquitination, which is involved with Parkin-independent, Drp1 mediation. Previous evidence has indicated that Drp1 can be activated in different ways to drive mitochondria division in mammals ^{2,3}. It stands to reason that the role of Drp1 in this

model of mitophagy may participate in providing mitochondrial fragments small enough to be cleared by autophagy⁴, but the detailed mechanism needs to be further investigated. Additionally, a preceding study has demonstrated that the autophagy adaptor SQSTM1 recruits two CRL subunits Keap1 and Rbx1 to the mitochondria and then promotes OMM ubiquitination⁵. While, *B. pseudomallei*-induced mitophagy is more likely to require SQSTM1 to interact with ubiquitinated IMMT, thereby recruiting LC3-bound autophagosomes to the mitochondria. Suggesting an explanation that IMMT could not function as a mitophagy receptor directly associated with LC3 in the absence of a predicted LIR motif'. Thanks.

4. In addition to showing evidence of mitochondrial inner membrane exposure and ubiquitination, using microscopy, can the authors assess whether autophagy machineries are recruited (e.g. ATG13, WIPI2), including ubiquitin binding autophagy adaptors that are known to drive mitophagy (e.g. NDP52, Optineurin)?

Answer: We appreciate the reviewer's constructive suggestions.

(a) According to the reviewer's suggestion, we have performed these relevant experiments in Extended Data Fig. 2 in the revised manuscript, and described detailedly above (question #1). Thanks again for your valuable suggestions.

(b) Based on the data of ubiquitin proteomics, SQSTM1, as an adaptor implicated in Ub-dependent mitophagy, was identified to show evidently increased ubiquitination under BipD overexpression compared to the control (Fig. 5e). Thus, we reasoned that SQSTM1 may be involved in the ubiquitinated IMMT triggered mitophagy. As shown in the Extended Data Fig. 9a in the revised manuscript, we found that SQSTM1 colocalized with Ub in *B. pseudomallei*-infected RAW264.7 cells. Similarly, the autophagosome marker LC3 was also observed to colocalize with Ub (Extended Data Fig. 9b in the revised manuscript). Collectively, this finding supports the possibility that ubiquitinated IMMT provides the signal for recruiting SQSTM1 upon *B. pseudomallei* infection, mediating the initiation of host mitophagy. The result is now shown in Extended Data Fig. 9a,b and we point this out in the revised manuscript on page 10 line 245-250. Thanks.

5. Is BipD and the associated ubiquitin ligase complex recruited to the exposed inner membrane of mitochondria (as assessed by microscopy)?

Answer: We thank the reviewer's constructive suggestions. We performed immunofluorescence staining to investigate whether BipD and CUL3 complex were recruited to mitochondrial HSP60 after *B. pseudomallei* Δ bipD infection. The result is now shown in Extended Data Fig. 8c and we point this out in the revised manuscript

on page 10 line 240-241 as “Next, we further confirmed that CUL3 were colocalized with mitochondrial HSP60 under ectopic expression of BipD by fluorescence microscopy”. Thanks.

6. Mdivi-1, originally developed as a Drp1 inhibitor, is a well established off-target compound and not a specific inhibitor of mitophagy (e.g. Bordt et al (2017) Dev Cell). Therefore, the experiments using this compound as an autophagy inhibitor are not highly informative and the authors would be better served not including data using Mdivi-1.

Answer: We thank the reviewer for pointing out this issue. As mentioned in Bordt EA et al. work, despite the compound mdivi-1 is widely reported to inhibit Drp1-dependent fission and elongate mitochondria ^{6,7}, they have found that mdivi-1 is more likely to be a relatively unusual complex I inhibitor that is not only weak and reversible but also has the ability to attenuate pathological ROS production at the complex I Q site, with limited impact on ROS in healthy neurons, especially in a Drp1-independent manner ⁸. Importantly, according to the reviewer’s valuable suggestions, we have determined that Drp-1 is likely to be involved in *B. pseudomallei*-induced mitophagy, but the structure of mdivi-1 contains a thiophenol and may have multiple cellular targets and effects ⁸. Therefore, we totally agree with the reviewer’s suggestion, it is not applicable to use mdivi-1 as a mitophagy inhibitor in our study, and we have deleted the relevant data of mdivi-1 treatment in the Fig. 2 in the revised manuscript. Thanks very much.

7. To date, the factors known to result in rupture of the outer membrane during inner membrane mediated mitophagy are Parkin (Yoshii et al (2011) J Biol Chem), Bax and Bak (McArthur et al (2018 Science), and Drp1 (Oshima et al (2021) J Cell Biol). The authors have excluded Parkin, can they also assess whether Drp1 or BAX/Bak contribute to the infection induced inner membrane mitophagy they observe?

Answer: We appreciate the reviewer’s valuable suggestions. According to the current evidence, the factors known to result in rupture of the outer membrane during inner membrane mediated mitophagy are Parkin ⁹, Bax/Bak ¹⁰, and Drp1 ². We have excluded the possibility of PINK/Parkin involvement. Therefore, we investigated the possibility whether the Bax/Bak or Drp1 was involved in *B. pseudomallei*-induced mitophagy. As shown in the Extended Data Fig. 9c-e in the revised manuscript, knockdown of Drp1 obviously reversed the decline of mtDNA upon *B. pseudomallei* infection, while Bax and Bak were not likely required for *B. pseudomallei*-induced

mitophagy. Moreover, we further observed that the colocalization of mitochondria with LC3 markedly decreased when Drp1 was knockdown (Extended Data Fig. 9f in the revised manuscript). Similar effect was also obtained in $\Delta\psi_m$ detection (Extended Data Fig. 9g in the revised manuscript). Collectively, these results indicate that Drp1 may be crucial for *B. pseudomallei*-induced mitophagy. The result is now shown in Extended Data Fig. 9c-g and we point this out in the revised manuscript on page 10 line 250-259. Thanks.

8. Can the authors utilize knockdown/knockout of IMMT and rescue with WT or K211A mutant IMMT followed by mtKeima mitophagy analysis to directly assess the requirement for IMMT K211 in mitophagy?

Answer: We appreciate the reviewer's constructive suggestions. Firstly, *IMMT*^{-/-} RAW264.7 cells stably expressing mt-Keima were transfected with overexpression plasmids of Flag-IMMT or Flag-IMMT^{K211R}, respectively, and then infected with *B. pseudomallei*. As shown in Fig. 7i in the revised manuscript, *B. pseudomallei* infection induced a distinct mitophagy signal in Flag-IMMT overexpression cells, while this alteration was abolished by substitution of K211 of IMMT. The result is now shown in Fig. 7i and we point this out in the revised manuscript on page 12 line 316-321 as "Besides, to further assess the requirement for IMMT K211 in mitophagy, mt-Keima assay was used to assess the mitophagy level induced by *B. pseudomallei* infection. As expected, *B. pseudomallei* infection induced a distinct mitophagy signal in Flag-IMMT overexpression cells (Fig. 7i). In contrast, this alteration was abolished by substitution of K211 of IMMT (Fig. 7i), indicating that the K211 of IMMT is required for *B. pseudomallei*-induced mitophagy occurrence". Thanks.

Reviewer #2 (Remarks to the Author):

This article uncovers something novel about the ability of *B. pseudomallei* to induce mitophagy to evade intracellular killing. It is well organized and presents several interesting pieces of data on the infection biology of the bacteria. However, there are several areas that require major clarifications and amendments.

1. The study did not address prior findings in the field on *B. pseudomallei*'s ability to induce autophagy and its interference in ubiquitination. Often, these studies are in contradiction to the current findings, and the authors could explain the discrepancies, eg due to differences in techniques and also the fact that the bacteria have several mechanisms in operation that are working in opposite directions. It depends on when all these effectors are expressed, and this point is often obscured when there is too high an MOI of infection or a careful kinetic study is not undertaken.

a) BopA has been shown to inhibit autophagy, and that has been shown to increase intracellular bacterial counts. However, the study did not examine whether the autophagy they observed in fact could be mitophagy instead. The authors should cite these papers PMID: 21412437, PMID: 18483470 and interact with these findings. Would a bopA mutant shows an opposite phenotype to bipD mutant in terms of mitophagy? Or the 2 are independent phenomena?

Answer: We thank the reviewer for pointing out this issue. According to the previous studies, both BopA and BipD have been identified to be important components of T3SS3, which is a crucial virulence factor in *B. pseudomallei*. Especially, Cullinane M et al. have discovered that BopA facilitates bacterial evasion of autophagy by disturbing co-localization of LC3 to *B. pseudomallei*¹¹. Besides, Gong L et al. further confirmed that BopA was identified to interfere with LC3-associated phagocytosis (LAP) in *B. pseudomallei*-infected RAW264.7 cells¹². Therefore, in order to determine whether BopA is involved in the BipD-mediated mitophagy initiation, we firstly measured the mitochondrial DNA and protein in *B. pseudomallei* WT or $\Delta bopA$ -infected RAW264.7 cells. As shown in Extended Data Fig. 7a,b in the revised manuscript, infection with *B. pseudomallei* WT and $\Delta bopA$ resulted in a similar decrease in the levels of mtDNA, HSP60 and TIM23, compared to the control. Then, we analyzed the alteration of $\Delta\psi_m$ after infection, and found there was no differences between WT and $\Delta bopA$ in the $\Delta\psi_m$ changes (Extended Data Fig. 7c in the revised manuscript), which indicates that it appears unlikely that BopA is required for mitophagy initiation in response to *B. pseudomallei* infection. In particular, the

detailed mechanism of BopA-mediated inhibition of LC3 recruitment to *B. pseudomallei*-containing phagosomes, a process designated LAP, currently remains unclear. As mentioned in Cullinane M et al. work, BopA contains a Rho GTPase inactivation domain at its carboxy terminus functioning as a possible protease or acyltransferase acting on host molecules, which needs a detailed investigation of its cellular location and biochemical function^{12,13}. Of note, our co-investigator Prof. Tao Peng (Peking University ShenZhen Graduate School) has recently uncovered that BopA seems like a potential enzyme involved in the palmitoylation modification of substrates, and further research is being carried out (data unpublished). In our study, we have found BipD promotes the clearance of damaged mitochondria by mitophagy and reduces mtROS production by hijacking the host KLHL9/KLHL13/CUL3 E3 ligase complex to ubiquitinate IMMT. Taken together, in our opinion, BopA and BipD are two important molecules, which are essential for *B. pseudomallei* to escape from the immune clearance of host defense response through different mechanisms. But the detailed mechanisms show high levels of complexity and need to be further investigated. We have added this description in the Results part 3 on page 8 line 186-191 and Discussion on page 14 line 364-372 in the revised manuscript, respectively. Thanks.

b) the authors also did not address previous study showing TssM PMID: 20335533 as a deubiquitinase. These factors are all in PMID: 29848489 bacteria. This could be addressed in the Discussion.

Answer: We thank the reviewer's kind comments. Signaling by ubiquitination regulates virtually every cellular process in eukaryotes. Given the essential roles of ubiquitination in the regulation of the immune system, it is not surprising that the ubiquitination process is the target of many bacterial pathogens, which have developed techniques to hijack it for their own benefit^{14,15}. Numerous studies have uncovered that pathogenic bacteria have developed the means to interfere extensively with different stages of ubiquitination and deubiquitination pathways¹⁶. According to the previous studies, Akturk A et al.¹⁷ Dong Y et al.¹⁸ and Kalayil S et al.¹⁹ describe the X-ray crystal structure of the *Legionella* effector SdeA, which catalyzes ubiquitination. And Wang Y et al. report the structure of a *Legionella* enzyme SidE from the same protein family as SdeA²⁰. Furthermore, *Salmonella* SseL acts as a deubiquitinating enzyme (DUB) *in vivo*^{21,22}, while *Salmonella* SopA, which is a HECT-like E3 ubiquitin ligase, can induce ubiquitination of bacterial and host proteins²³; *Shigella* OspG could prevent phospho-IκBα degradation by targeting Ub-conjugating enzymes²⁴, while *Shigella* IpaH9.8 acts as an E3 ubiquitin ligase to promote ubiquitination of its substrate *in vivo*²⁵. Besides, SARS-CoV-2 infection can

also lead to changes in the ubiquitination system, affecting E3 ubiquitin ligases, DUBs, and enzymes involved in modifying and removing ubiquitin-like molecules ²⁶. For example, SARS-CoV-2 S protein can inhibit the ubiquitination of ACE2 and promote virus invasion, Nsp5 acts as an E3 ubiquitin ligases and mediates the K63 ubiquitination of MAVS on the K136 site, and PLpro acts as a DUBs *in vivo* ^{27, 28, 29, 30}.

Tan KS et al. have revealed that *B. pseudomallei* TssM, which is the homologous protein of *B. mallei* TssM, acts as a DUBs ³¹. Although the *B. mallei* TssM has been shown to extensively deubiquitinate cellular proteins *in vitro*, its cellular substrates or phenotype either in cellular assays or animal models have not been found ³². Tan KS et al. have identified that *B. pseudomallei* TssM inhibits the activation of NF- κ B by promoting its substrates TRAF-6, I κ B α and TRAF-3 deubiquitination, and then facilitates intracellular survival of *B. pseudomallei* ³¹. In our study, we found that *B. pseudomallei* BipD hijacks host KLHL9/KLHL13/CUL3 E3 ligase complex to ubiquitinate the mitochondrial substrate IMMT, that initiates mitophagy to evade killing. These findings have indicated that bacteria have developed mechanisms to evade or exploit the processes activated by ubiquitination, producing both ubiquitin ligases and deubiquitinates that modulate host defense responses. We have added this description in the Discussion on page 15 line 384-398 in the revised manuscript. Thanks.

c) Cif is another factor of Bp, although it is not present in all strains of Bp. It can increase slightly intracellular counts and partly through inhibition of Cullin RING E3 ubiquitin ligase PMID: 29848489

Answer: We appreciate the reviewer for pointing out this issue. Indeed, Cui JX et al. have reported that Cif and CHBP (Cif homolog from *B. pseudomallei*) can catalyze the deamidation of Gln⁴⁰ in ubiquitin and NEDD8 ³³. In cellular assays, they have found inhibition of CRL activity by Cif results in the stabilization of CRL substrates in EPEC-infected HeLa cells, including the cell cycle inhibitors Nrf2 and p27 ³³. Of note, they also have confirmed CHBP deamidates Gln⁴⁰ in ubiquitin and NEDD8 *in vitro* activity assay, and during *B. thailandensis* infection harboring a CHBP expression plasmid ³³. Furthermore, Ng MY et al. have found that CHBP could activate MAPK/ERK signaling and exert a bimodal effect on NF- κ B signaling, which is dependent on its deamidase activity, but independent of Cullin E3 ligase inhibition ^{34, 35}. These findings indicated that besides CRL inhibition, CHBP has additional cellular effects. They also found CHBP transfection at high levels had only a moderate effect on *B. thailandensis* intracellular replication, suggesting that CHBP plays only a minor role as a virulence factor ³⁵. Of note, many pathogenic *B. pseudomallei* strains lack the *chbp* gene, further suggesting that CHBP is not an

essential virulence factor in *B. pseudomallei*³⁵. Therefore, whether CHBP-mediated CRL inhibition is of importance during *B. pseudomallei in vivo* infection remains to be determined, and the level or timing of CHBP secretion is also required to be further characterized. In summary, according to the above discoveries and our current results, it is demonstrated that *B. pseudomallei* has evolved a large array of mechanisms to survive and replicate within eukaryotic cells, which still remains to be further defined. We have added this description in the Discussion on page 15 line 384-398 in the revised manuscript. Thanks.

2. Is the *bipD* mutant still able to secrete *bsaT3SS* effectors? Or that it is in fact causing over secretion of effectors? If the *BipD* mutation alters the secretion of effectors, including those that may have an opposite effect like *BopA*, then using the mutant to validate the infection and intracellular survival is problematic. Do the authors know the domain of *BipD* that interacts with the *Cul3* complex? Through site directed mutagenesis, the authors could determine which mutations allow *BipD* to be translocated but it is now unable to interact. If this mutant sequence is complemented into the *bipD* mutant, and the ability to induce mitophagy and intracellular counts are measured, it will increase the validity of the claim.

Answer: We appreciate the reviewer's constructive suggestions. After investigating literature, there is not much research has been reported about what effect *bipD* mutant may have on the secretion of T3SS3 effectors. Gong L et al. have mentioned that *bipD* mutant might be unable to secrete any effector proteins during *B. pseudomallei* infection¹². In addition, Broek CW et al. have found that the deletion of *bipD* was observed to hypersecrete the known Bsa effectors into the bacterial culture supernatant such as BopE, BapC, BopA etc., and two new substrates by iTRAQ technique^{36,37}. Generally, it is accepted that Type III secretion is a tightly regulated process initiated only after host cell contact, in order to ensure substrates secreted in the correct order³⁸. Both sequence and structure of BipD and *Shigella* IpaD showed high similarity, suggesting BipD may have a similar effect as IpaD. *Shigella* IpaD is the needle cap protein, acting to block secretion of effector proteins until host cell contact has taken place³⁹. Thus, under the bacterial culture condition, deletion of *ipaD* creates a "leaky" phenotype, in which higher levels of both translocators and effectors are seen in the supernatant⁴⁰. While, under infection conditions, the secretion signal has been activated, and then secretion happens in an ordered way^{41,42}, so it is not sure how *ipaD* mutant plays a role in the secretion of translocators and effectors during *Shigella* infection. Besides, we still do not ascertain how the activation signal generated after contact with the host cell is transduced to the cytoplasmic side of the T3SS to initiate secretion. The first part of the T3SS that

establishes contact with the host cell is probably the needle tip. Therefore, it is likely that the tip complex senses the host cell. Based on the above analysis, considering its high complexity on the regulation of the T3SSs, it is still not well understood that what effect *bipD* deletion may have on the secretion of T3SS3 effectors and translocators when *B. pseudomallei* infected host cells, which needs further detailed investigation.

Erskine PT et al. have revealed the crystal structure of BipD solved at a resolution of 2.1 Å, and identified the core four-helix bundle of BipD is formed by the helices 2, 3, 4 and 8. However, we still do not understand the functional domain or particular site of BipD involved in translocation and secretion, which may need a random mutagenesis with a genetic screen to identification and is full of challenges. Besides, in order to determine the domain of BipD that interacts with the CUL3 complex, we tried to perform an interaction analysis based on molecular structure docking GRAMM Web ⁴³ (<https://gramm.compbio.ku.edu/results/31208/0>) and PDBePISA (https://www.ebi.ac.uk/msd-srv/prot_int/pistart.html) to predict the potential key sites for BipD to interact with KLHL9 and KLHL13, respectively. As shown in Figure 1a,b below, rigid docking of BipD/KLHL9 and BipD/KLHL13 was performed, and found that there were multiple amino acids of BipD could form hydrogen bonds with KLHL9 and KLHL13, respectively. Among these sites, the hydrogen bonds with the bond lengths less than 3 Å between BipD and KLHL9 are likely to located on the THR 200, GLN 202, ARG 97, ASP 145, ARG 100 and ILE 222 of BipD, while the hydrogen bonds with the bond lengths less than 3 Å between BipD and KLHL13 are likely to located on the ASP 224 and SER 225 of BipD. Additionally, there were other 8 or 6 amino acid sites that may form hydrogen bonds with KLHL9 or KLHL13, respectively (Figure 1c,d below). However, the detailed form of BipD interaction with KLHL9 and KLHL13 is still intricate and complex, and the key sites for the translocation and secretion of BipD was still unclear. Thus, the subsequent screening and verification of these potential amino acids would be complicated and time-consuming.

Therefore, we have to determine to perform alternative experiments to verify the functions of BipD and BopA as following. Firstly, ectopic expression of BipD in cells infected with Δ *bipD* strain could rescue the decrease in the mtDNA, and promote the clearance of accumulated mitochondrial proteins, while expression of BopA showed no differences under Δ *bopA* infection (Extended Data Fig. 7d,e in the revised manuscript). Similar effect was observed in the colocalization of mitochondria marker HSP60 and LC3 (Extended Data Fig. 7f,g in the revised manuscript). Taken together, these results have suggested that BipD is requires for *B. pseudomallei*-induced mitophagy, while BopA may not be involved in this process. We have added this description in the Results part 3 in the revised manuscript on page 8 line 191-197.

Thanks.

Figure 1

(a and b) Predicted binding modes of KLHL9 (a) and KLHL13 (b) substrate to BipD according to the molecular structure rigid docking by GRAMM Web.

(c and d) Potential amino acid sites of KLHL9 and BipD (c) or KLHL13 and BipD (d) that may form the hydrogen bonds predicted by GRAMM Web.

3. BipD is also present in *B. thailandensis* but the authors claim that this doesn't happen in *Bt* infection. However, this claim may be wrong as *Bt* infection follows a slower kinetics. If the authors measure the effects with longer time points, they may see the same thing. If not, the authors could examine the protein sequence differences between BipD of the 2 species.

Answer: We thank the reviewer's kind comments. Firstly, we reorganized to measure the levels of $\Delta\psi_m$ and mtDNA in RAW264.7 cells after *B. thailandensis* infection for a longer infection time (6, 10, 14 hpi) at an MOI of 10, and found there was no significant changes in alterations of $\Delta\psi_m$ and mtDNA after *B. thailandensis* infection, which suggesting *B. thailandensis* infection may not induce the mitophagy in mouse macrophages (Figure 2a,b below). Then, we compared the protein sequence differences between BipD of the 2 species by sequence alignment, and exhibited 90.6% amino acid identity and 93.5% similarity (Figure 2c below). Particularly, in our study, we found that BipD alone could not induce mitophagy without mitochondria damaged stimuli, except by CCCP treatment or *B. pseudomallei* infection. Our data suggested that the existence of bacterial proteins or toxins led to mitochondria

damaged during *B. pseudomallei* infection, which needs to be further investigated. Based on these results, it is suggested that *B. thailandensis* may be insufficient to trigger obvious mitochondrial damage during our observed infection, although we could not exclude the possibility that BipD of *B. thailandensis* could promote mitophagy in a similar to BipD of *B. pseudomallei*. Therefore, combined with the reviewer's suggestion, we think that this part of results is not very relevant to support our findings. So, we have deleted this part of results in the revised manuscript. Thanks.

Figure 2

(a) Quantification of mtDNA/nDNA levels in RAW264.7 cells infected with *B. thailandensis* for (0, 6, 10, 14 h) at an MOI of 10 by qPCR analysis. hpi, hours post infection.

(b) Quantification of TMRM fluorescence intensity of the RAW264.7 cells infected with *B. thailandensis* by flow cytometry analysis. Data was from 3 independent experiments, and showed means \pm SD (one-way ANOVA).

(c) Protein sequence alignment between BipD of *B. pseudomallei* and *B. thailandensis*.

4. Figure 7. It helps if the authors could use densitometry to quantify their western blots where normalization to control proteins are shown. Figure 7d shows a different coloration on the strip on IMMT under IMMT- lanes. Please state either the 3 lanes were not done or to clarify.

Answer: We thank the reviewer for pointing out this issue and totally agree with your suggestion. Firstly, we have used densitometry to quantify western blots data in the revised manuscript. Secondly, we are so sorry that we did not show the IMMT lane of

the Fig. 7d in a good way. The different coloration on the strip on IMMT under IMMT^{-/-} lanes may be due to the cropping of the strip image. As the original image without cropping which we have already uploaded to Mendeley Data (https://data.mendeley.com/?dgcid=em-letter_editorial_mendeleydata-cellpress-submission, doi: 10.17632/v4xny7kfd8.1) (Figure 3a below), the coloration on the strip on IMMT under IMMT^{-/-} lanes is basically the same. In addition, the data we showed in the first manuscript were from one of 3 independent experiments, and another two results are shown in the Figure 3b,c below. Therefore, another representative data has been shown in the Fig. 7d in the revised manuscript. Thanks.

Figure 3

(a) The original image of Fig. 7d.

(b and c) Another two results for testing the effect of IMMT on *B. pseudomallei* infection induced mitophagy.

References:

1. Yamamoto H, Zhang S, Mizushima N. Autophagy genes in biology and disease. *Nature reviews Genetics* **24**, 382-400 (2023).
2. Oshima Y, *et al.* Parkin-independent mitophagy via Drp1-mediated outer membrane severing and inner membrane ubiquitination. *J Cell Biol* **220**, (2021).
3. Kleele T, *et al.* Distinct fission signatures predict mitochondrial degradation or biogenesis. *Nature* **593**, 435-439 (2021).
4. Karbowski M, Oshima Y, Verhoeven N. Mitochondrial proteotoxicity: implications and ubiquitin-dependent quality control mechanisms. *Cell Mol Life Sci* **79**, 574 (2022).
5. Yamada T, *et al.* Mitochondrial Stasis Reveals p62-Mediated Ubiquitination in Parkin-Independent Mitophagy and Mitigates Nonalcoholic Fatty Liver Disease. *Cell Metab* **28**, 588-604 e585 (2018).

6. Cassidy-Stone A, *et al.* Chemical inhibition of the mitochondrial division dynamin reveals its role in Bax/Bak-dependent mitochondrial outer membrane permeabilization. *Dev Cell* **14**, 193-204 (2008).
7. Rosdah AA, J KH, Delbridge LM, Dusting GJ, Lim SY. Mitochondrial fission - a drug target for cytoprotection or cytodestruction? *Pharmacol Res Perspect* **4**, e00235 (2016).
8. Bordt EA, *et al.* The Putative Drp1 Inhibitor mdivi-1 Is a Reversible Mitochondrial Complex I Inhibitor that Modulates Reactive Oxygen Species. *Dev Cell* **40**, 583-594 e586 (2017).
9. Yoshii SR, Kishi C, Ishihara N, Mizushima N. Parkin mediates proteasome-dependent protein degradation and rupture of the outer mitochondrial membrane. *J Biol Chem* **286**, 19630-19640 (2011).
10. McArthur K, *et al.* BAK/BAX macropores facilitate mitochondrial herniation and mtDNA efflux during apoptosis. *Science* **359**, (2018).
11. Cullinane M, *et al.* Stimulation of autophagy suppresses the intracellular survival of *Burkholderia pseudomallei* in mammalian cell lines. *Autophagy* **4**, 744-753 (2008).
12. Gong L, *et al.* The *Burkholderia pseudomallei* type III secretion system and BopA are required for evasion of LC3-associated phagocytosis. *PLoS One* **6**, e17852 (2011).
13. Pei J, Grishin NV. The Rho GTPase inactivation domain in *Vibrio cholerae* MARTX toxin has a circularly permuted papain-like thiol protease fold. *Proteins* **77**, 413-419 (2009).
14. Trulsson F, *et al.* Deubiquitinating enzymes and the proteasome regulate preferential sets of ubiquitin substrates. *Nat Commun* **13**, 2736 (2022).
15. Kubori T, Kitao T, Nagai H. Emerging insights into bacterial deubiquitinases. *Curr Opin Microbiol* **47**, 14-19 (2019).
16. Rytkonen A, Holden DW. Bacterial interference of ubiquitination and deubiquitination. *Cell Host Microbe* **1**, 13-22 (2007).
17. Akturk A, *et al.* Mechanism of phosphoribosyl-ubiquitination mediated by a single *Legionella* effector. *Nature* **557**, 729-733 (2018).
18. Dong Y, *et al.* Structural basis of ubiquitin modification by the *Legionella* effector SdeA. *Nature* **557**, 674-678 (2018).
19. Kalayil S, *et al.* Insights into catalysis and function of phosphoribosyl-linked serine

- ubiquitination. *Nature* **557**, 734-+ (2018).
20. Wang Y, *et al.* Structural Insights into Non-canonical Ubiquitination Catalyzed by SidE. *Cell* **173**, 1231-+ (2018).
 21. Le Negrate G, *et al.* Salmonella secreted factor L deubiquitinase of Salmonella typhimurium inhibits NF-kappaB, suppresses IkappaBalpha ubiquitination and modulates innate immune responses. *Journal of immunology (Baltimore, Md : 1950)* **180**, 5045-5056 (2008).
 22. Rytkönen A, *et al.* SseL, a Salmonella deubiquitinase required for macrophage killing and virulence. *Proceedings of the National Academy of Sciences of the United States of America* **104**, 3502-3507 (2007).
 23. Zhang Y, Higashide WM, McCormick BA, Chen J, Zhou D. The inflammation-associated Salmonella SopA is a HECT-like E3 ubiquitin ligase. *Molecular microbiology* **62**, 786-793 (2006).
 24. Kim DW, Lenzen G, Page AL, Legrain P, Sansonetti PJ, Parsot C. The Shigella flexneri effector OspG interferes with innate immune responses by targeting ubiquitin-conjugating enzymes. *Proceedings of the National Academy of Sciences of the United States of America* **102**, 14046-14051 (2005).
 25. Rohde JR, Breitkreutz A, Chenal A, Sansonetti PJ, Parsot C. Type III secretion effectors of the IpaH family are E3 ubiquitin ligases. *Cell Host Microbe* **1**, 77-83 (2007).
 26. Zhao M, Zhang M, Yang Z, Zhou Z, Huang J, Zhao B. Role of E3 ubiquitin ligases and deubiquitinating enzymes in SARS-CoV-2 infection. *Front Cell Infect Microbiol* **13**, 1217383 (2023).
 27. Freitas BT, *et al.* Characterization and Noncovalent Inhibition of the Deubiquitinase and deISGylase Activity of SARS-CoV-2 Papain-Like Protease. *ACS infectious diseases* **6**, 2099-2109 (2020).
 28. Liu Y, *et al.* SARS-CoV-2 Nsp5 Demonstrates Two Distinct Mechanisms Targeting RIG-I and MAVS To Evade the Innate Immune Response. *mBio* **12**, e0233521 (2021).
 29. Freitas BT, *et al.* Exploring Noncovalent Protease Inhibitors for the Treatment of Severe Acute Respiratory Syndrome and Severe Acute Respiratory Syndrome-Like Coronaviruses. *ACS infectious diseases* **8**, 596-611 (2022).
 30. Klemm T, *et al.* Mechanism and inhibition of the papain-like protease, PLpro, of SARS-CoV-2. *The EMBO journal* **39**, e106275 (2020).

31. Tan KS, *et al.* Suppression of host innate immune response by Burkholderia pseudomallei through the virulence factor TssM. *Journal of immunology (Baltimore, Md : 1950)* **184**, 5160-5171 (2010).
32. Shanks J, *et al.* Burkholderia mallei tssM encodes a putative deubiquitinase that is secreted and expressed inside infected RAW 264.7 murine macrophages. *Infection and immunity* **77**, 1636-1648 (2009).
33. Cui J, *et al.* Glutamine deamidation and dysfunction of ubiquitin/NEDD8 induced by a bacterial effector family. *Science* **329**, 1215-1218 (2010).
34. Ng MY, Wang M, Casey PJ, Gan YH, Hagen T. Activation of MAPK/ERK signaling by Burkholderia pseudomallei cycle inhibiting factor (Cif). *PLoS One* **12**, e0171464 (2017).
35. Ng MY, Gan YH, Hagen T. Characterisation of cellular effects of Burkholderia pseudomallei cycle inhibiting factor (Cif). *Biology open* **7**, (2018).
36. Stevens MP, *et al.* A Burkholderia pseudomallei type III secreted protein, BopE, facilitates bacterial invasion of epithelial cells and exhibits guanine nucleotide exchange factor activity. *J Bacteriol* **185**, 4992-4996 (2003).
37. Vander Broek CW, Chalmers KJ, Stevens MP, Stevens JM. Quantitative Proteomic Analysis of Burkholderia pseudomallei Bsa Type III Secretion System Effectors Using Hypersecreting Mutants. *Mol Cell Proteomics* **14**, 905-916 (2015).
38. Buttner D. Protein export according to schedule: architecture, assembly, and regulation of type III secretion systems from plant- and animal-pathogenic bacteria. *Microbiol Mol Biol Rev* **76**, 262-310 (2012).
39. Roehrich AD, Guillosoou E, Blocker AJ, Martinez-Argudo I. Shigella IpaD has a dual role: signal transduction from the type III secretion system needle tip and intracellular secretion regulation. *Molecular microbiology* **87**, 690-706 (2013).
40. Picking WL, *et al.* IpaD of Shigella flexneri is independently required for regulation of Ipa protein secretion and efficient insertion of IpaB and IpaC into host membranes. *Infection and immunity* **73**, 1432-1440 (2005).
41. Kenjale R, *et al.* The needle component of the type III secretion of Shigella regulates the activity of the secretion apparatus. *J Biol Chem* **280**, 42929-42937 (2005).
42. Martinez-Argudo I, Blocker AJ. The Shigella T3SS needle transmits a signal for MxiC release, which controls secretion of effectors. *Molecular microbiology* **78**, 1365-1378 (2010).

43. Singh A, Copeland MM, Kundrotas PJ, Vakser IA. GRAMM Web Server for Protein Docking. *Methods in molecular biology (Clifton, NJ)* **2714**, 101-112 (2024).

REVIEWER COMMENTS

Reviewer #2 (Remarks to the Author):

The authors carefully answered all my queries in their rebuttal and revised manuscript. They also have included additional experiments that further clarify the mechanisms of bacterial BipD's direct role in the mitophagy process.

Recommend publication.

Reviewer #3 (Remarks to the Author):

Up to Reviewer comment #4 the authors adequately addressed the valid reviewer concerns. (some revision experiments are good and convincing – others such as unquantified EM image examples not very compelling but seem to be an acceptable standard these days). For reviewer comment #4 the authors include “colocalization data” in Extended Fig. 9a. My visual examination of the images does not reveal colocalization. The authors need to quantify scores of cells and use Pearson's correlation coefficient to determine if SQSTM1 and Ub colocalize. The same point holds for comment #5 and the reviewers response with Extended Fig. 8c where only one cell is shown – this needs to be quantified over scores of cells with use Pearson's correlation coefficient. The last issue that needs correction is the knock down of Bax and Bak in extended Fig. 9c-e. The minor decrease in the western blot in Bax and Bak is not sufficient to make any conclusion. Better knock downs or use of knock out cells (single and double KOs in HCT116 cells are available) is needed here. The results provided regarding Reviewer point #8 are convincing.

Reply to the comments from the Board of Editors and Reviewers:

Reviewer #2 (Remarks to the Author):

The authors carefully answered all my queries in their rebuttal and revised manuscript. They also have included additional experiments that further clarify the mechanisms of bacterial BipD's direct role in the mitophagy process. Recommend publication.

Answer: We thank the reviewer's constructive suggestions for improving our manuscript.

Reviewer #3 (Remarks to the Author):

1. Up to Reviewer comment #4 the authors adequately addressed the valid reviewer concerns. (some revision experiments are good and convincing – others such as unquantified EM image examples not very compelling but seem to be an acceptable standard these days). For reviewer comment #4 the authors include “colocalization data” in Extended Fig. 9a. My visual examination of the images does not reveal colocalization. The authors need to quantify scores of cells and use Pearson's correlation coefficient to determine if SQSTM1 and Ub colocalize. The same point holds for comment #5 and the reviewers response with Extended Fig. 8c where only one cell is shown – this needs to be quantified over scores of cells with use Pearson's correlation coefficient.

Answer: We thank the reviewer's understanding and appreciate the constructive suggestions. As shown in the Extended Fig. 8c in the revised manuscript, colocalization of CUL3 with mitochondrial HSP60 was analyzed by Pearson's correlation coefficient, which revealed a significant increase in the colocalization of CUL3 and HSP60 under BipD overexpression. In addition, as shown in the Extended Fig. 9a,b in the revised manuscript, fluorescence intensity profiles are shown on the right of images, and quantification of positions of SQSTM1 and LC3 relative to Ub signal was measured in the Extended Fig. 9c in the revised manuscript. Thanks.

2. The last issue that needs correction is the knock down of Bax and Bak in extended Fig. 9c-e. The minor decrease in the western blot in Bax and Bak is not sufficient to make any conclusion. Better knock downs or use of knock out cells (single and double KOs in HCT116 cells are available) is needed here.

Answer: We appreciate the reviewer's constructive suggestions. We performed this experiment using other three siRNAs targeting Bax and Bak, respectively. And according to knock down efficiency, we selected the best one (marked grey) for subsequent experiments (Table 1 below). As shown in the Extended Data Fig. 9d-f in

the revised manuscript, similar effect was also obtained in the alteration of mtDNA upon *B. pseudomallei* infection, suggesting that Bax and Bak were not likely required for *B. pseudomallei*-induced mitophagy. Thanks.

Table 1. siRNAs targeting Bax and Bak used in this study.

Previously used in this study	Currently used in this study
siRNAs targeting Bax AGGCAUUUUUCUUACUUUUTT	siRNAs targeting Bax #1 UAUGGAGCUGCAGAGGAUGTT (selected)
siRNAs targeting Bak ACCGUAAUGGUGAUUUUUGTT	#2 CUGUGUCUUUUCUUCAUAATT #3 UUCUUCAUAAAUAUGACATT siRNAs targeting Bak #1 UGCCUACGAACUCUUCACCTT (selected)
	#2 AGGUCUUUCGAAGCUACGUTT #3 CUGCUUUUUGGCUGAUUAUCTT

3. The results provided regarding Reviewer point #8 are convincing.

Answer: We thank the reviewer's comment.

REVIEWERS' COMMENTS

Reviewer #3 (Remarks to the Author):

The authors have adequately addressed the remaining concerns.

Point by point response for the third round of peer review:

Reply to the comments from the Board of Editors and Reviewers:

Reviewer #3 (Remarks to the Author):

The authors have adequately addressed the remaining concerns.

Answer: We thank the reviewer's constructive suggestions for improving our manuscript.